# DKD: DIRECTIONAL KNOWLEDGE DISTILLATION FOR ONE-STEP TEXT-TO-IMAGE GENERATION

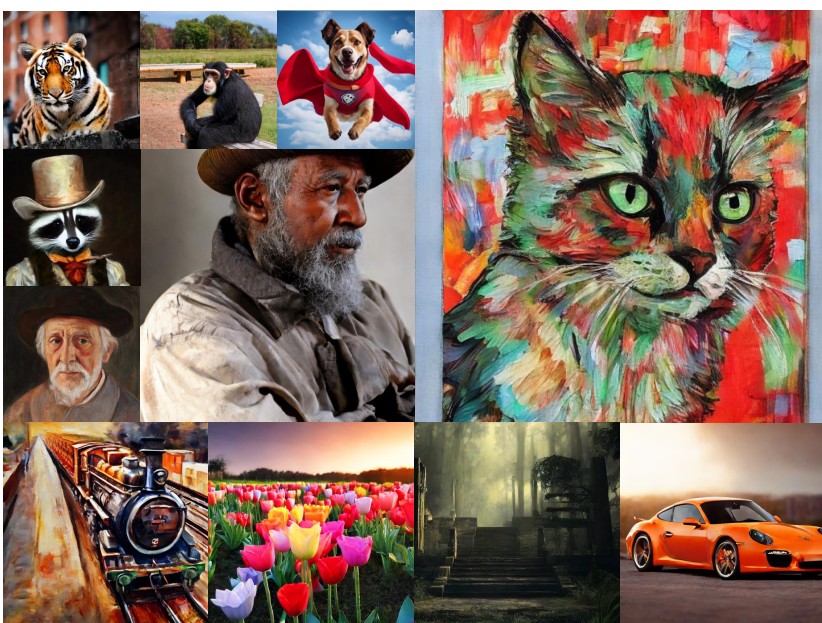

Figure 1: One-step generated images using our proposed method DKD (*i.e.*, SD 2.1).

## ABSTRACT

Despite the impressive performance of diffusion models such as Stable Diffusion (SD) in image generation, their slow inference limits practical deployment. Recent works accelerate inference by distilling multi-step diffusion into one-step generators. To better understand the distillation mechanism, we analyze U-Net/DiT weight changes between one-step students and their multi-step teacher counterparts. Our analysis reveals that changes in weight direction significantly exceed those in weight norm, highlighting it as the key factor during distillation. Motivated by this insight, we propose the **Lo**w-rank **R**otation of weight **D**irection (LoRaD). LoRaD is designed to model these structured directional changes using learnable low-rank rotation matrices. We further integrate LoRaD into Variational Score Distillation (VSD), resulting in Directional Knowledge Distillation (DKD)—a novel one-step distillation framework. DKD achieves state-of-the-art FID scores on COCO 2014 and COCO 2017 while using only approximately 10% of the trainable parameters of the U-Net. Furthermore, the distilled one-step model demonstrates strong versatility and scalability, generalizing well to various downstream tasks such as controllable generation, relation inversion, and high-resolution synthesis.

## 1 INTRODUCTION

Diffusion models (DMs) (Ho et al., 2020; Sohl-Dickstein et al., 2015; Song et al., 2021b) have received considerable attention for their ability to generate high-quality and diverse content. Thus, they are widely applied to tasks such as text-to-image (Rombach et al., 2022; Li et al., 2024b; Ruiz et al., 2023; Zhang et al., 2023) generation, text-to-video (Khachatryan et al., 2023; Wu et al., 2023a; Zhou et al., 2024c; Kong et al., 2024) generation, and image-to-video (Wang et al., 2025; Ni et al., 2023; Bar-Tal et al., 2024; Hu et al., 2025) generation. However, the reliance of DMs on multiple

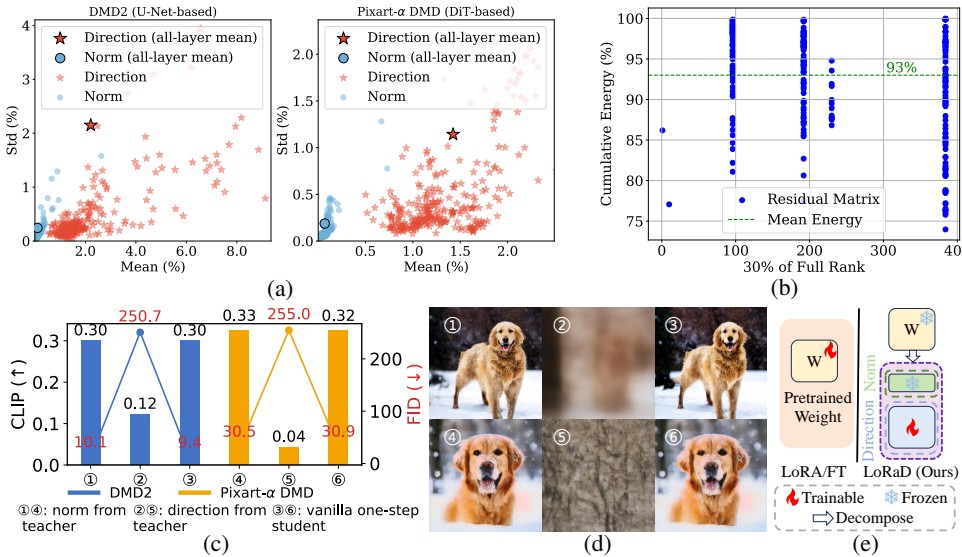

Figure 2: Motivational analysis of our method. (a) Differences in weight norm and direction between the one-step student and the teacher model. See Appendix E for details and additional examples. (b) SVD analysis of the residual matrix for DMD2. (c) Replacing the one-step model's norm with that of the multi-step model has little effect (①, ④); replacing the direction severely degrades generation quality (②, ⑤). (d) Qualitative examples corresponding to (b). (e) Illustration of LoRaD.

sampling steps leads to high computational cost and slow inference. To address this, recent distillation methods reduce the number of steps to a few (Luo et al., 2023b; Chadebec et al., 2025) or even one (Ren et al., 2024; Lin et al., 2024; Dao et al., 2024). Interestingly, during distillation we find the weight norm remains relatively small across layers, while the direction shows larger variations when reparameterizing weights into *norm* and *direction* for both teacher and student generators.

Inspired by the weight reparameterization (Salimans & Kingma, 2016; Liu et al., 2024), we adopt a similar decomposition to analyze weight changes in diffusion distillation. To begin our analysis, we examine weight updates between state-of-the-art (SOTA) one-step models (*e.g.*, DMD2 (Yin et al., 2024a) and Pixart-$\alpha$ DMD (Yin et al., 2024b)) and their corresponding multi-step counterparts (*e.g.*, SD 1.5 (Rombach et al., 2022) and Pixart-$\alpha$ (Chen et al., 2023)). As shown in Fig. 2 (a) (left), in U-Net–based architectures the weight norm remains nearly stable across layers, with a mean and standard deviation (STD) of 0.1% and 0.2%, respectively. In contrast, the weight direction exhibits much more pronounced change, with a mean of 2.2% and STD of 2.1%, corresponding to ratios of 22× and 10× those of the norm. A similar trend is observed in DiT–based architectures (see Fig. 2 (a) (right)). These observations suggest that the weight direction may carry richer and more sensitive information than the norm in distillation. Further, if the direction indeed accounts for the primary information differences, we ask whether these differences exhibit a structured pattern. To this end, we perform SVD on the residual matrix—the difference between the one-step and multi-step direction matrices—and find that retaining 30% of its rank recovers 93% of the information, highlighting its low-rank nature (see Fig. 2 (b)).

To quantify the impact of these two components, we conduct a controlled ablation study by selectively replacing either the norm or direction of the one-step model with that from the multi-step teacher (see Fig. 2 (d)). As shown in Fig. 2 (c), substituting the norm leads to negligible performance change (e.g., DMD2: +0.7 FID, unchanged CLIP), whereas substituting the directions causes severe degradation (e.g., DMD2: +241.3 FID, -0.18 CLIP). These findings suggest that the weight direction plays a primary role in distillation, while variation in the norm appears comparatively minor. One possible explanation is that initializing the student with teacher weights aligns the initial norm, and weight decay during training further constrains norm drift (Loshchilov & Hutter, 2017); the distillation signal then acts mainly through adjustments in the weight direction to reduce representational discrepancy (Salimans & Kingma, 2016). Taken together, these results indicate that ***direction reconstruction a key factor underlying performance improvement in distillation.***

The distillation methods mentioned above can be broadly categorized into two types: full fine-tuning (FT) and Low-Rank Adaptation (LoRA) (Hu et al., 2022)-based fine-tuning. However, they directly

update the model parameters while optimizing both norm and direction. The changes in norm and direction differ, with norm showing minimal variation and directions experiencing significant changes, which increases the optimization difficulty due to the strong coupling between them. Furthermore, both FT and LoRA face issues of slow convergence (Huang et al., 2024a; Dong et al., 2024), instability (Han et al., 2024; Hayou et al., 2024), and overfitting (Aghajanyan et al., 2021; Huang et al., 2025), further complicating the optimization process.

To address the above challenges, we propose Low-rank Rotation of weight Direction (LoRaD) (see Fig. 2 (e)), which adjusts the direction of pre-trained weights via learnable rotation matrices. Given the structured nature (*i.e.*, low-rank property) of directional changes, the rotation angles are parameterized as the product of two low-rank matrices to further reduce the number of learnable parameters. We integrate LoRaD into Variational Score Distillation (VSD) (Wang et al., 2023) and introduce Directional Knowledge Distillation (DKD), a novel one-step text-to-image distillation framework. Experiments on the COCO 2014 (Lin et al., 2014) and COCO 2017 (Lin et al., 2014) datasets show that DKD achieves SOTA FID scores, outperforming all existing one-step generation methods, This was accomplished by optimizing only the direction, which reduced the difficulty of distillation, while using only about **10%** of the U-Net parameters as trainable components—greatly enhancing parameter efficiency. Furthermore, we apply DKD to downstream tasks including controllable generation, relation inversion, high-resolution synthesis, and image customization, demonstrating its acceleration capability and broad applicability. Our contributions are summarized as follows:

- We conduct an in-depth analysis of weight changes in U-Net between multi-step and one-step generation models, which points to weight-direction adjustment as a key driver of one-step distillation. This provides a new theoretical perspective for efficient distillation.

- We propose a novel distillation framework for one-step text-to-image generation, named DKD, which employs LoRaD to model weight directions via low-rank rotations, effectively guiding the student model to align with the teacher distribution.

- DKD is evaluated on the COCO dataset and several downstream tasks. Both qualitative and quantitative results demonstrate that DKD significantly improves inference efficiency while achieving substantial gains in image quality.

## 2 RELATED WORK

**Diffusion models.** Diffusion models (Ho et al., 2020; Sohl-Dickstein et al., 2015; Song & Ermon, 2019; Song et al., 2021b) excel in image generation, but pixel-space computation imposes a heavy computational burden. To improve efficiency, Rombach et al. (2022) introduced Latent Diffusion Models (LDM), shifting denoising to latent space. However, existing text-guided methods (Rombach et al., 2022; Podell et al., 2023; Li et al., 2024b; Ruiz et al., 2023; Zhang et al., 2023) are still slow due to multi-step generation. While most use a U-Net backbone, Diffusion Transformer (DiT) (Peebles & Xie, 2023) replaces it with a Transformer for better scalability, advancing text-to-image generation (Chen et al., 2023; 2024b;a; Esser et al., 2024). Despite improvements, iterative denoising remains a slow process. Recently, many acceleration methods have emerged.

**Diffusion model acceleration.** The existing acceleration methods can be divided into training-free and training-based approaches. *Training-free acceleration methods* for diffusion models fall into two main categories. The first method, which reduces redundant computation through caching (Ma et al., 2024; Wimbauer et al., 2024; Selvaraju et al., 2024; Li et al., 2024a), is exemplified by Faster Diffusion (Li et al., 2024a). The second method uses high-order solvers (Song et al., 2021a; Liu et al., 2022; Zhang & Chen, 2022; Lu et al., 2022a;b), such as DDIM (Song et al., 2021a) and DPM-Solver (Lu et al., 2022a;b), to reduce the number of sampling steps. However, the acceleration effects of these two methods are limited, so training-based methods have received more attention.

*Training-based acceleration methods* can be broadly categorized into four groups: consistency distillation (CD), progressive distillation (PD), diffusion-GAN distillation, and variational score distillation (VSD). CD (Song et al., 2023; Wang et al., 2024; Ren et al., 2024; Kim et al., 2023; Luo et al., 2023a;b) learns trajectory-level consistency for faster sampling but often suffers from low image fidelity. PD (Salimans & Ho, 2022; Ren et al., 2024) reduces steps in stages, introducing significant training overhead. Diffusion-GAN distillation (Luo et al., 2024; Lin et al., 2024; Xu et al., 2024; Kang et al., 2024), such as Diffusion2GAN (Kang et al., 2024), enhances fidelity by distilling

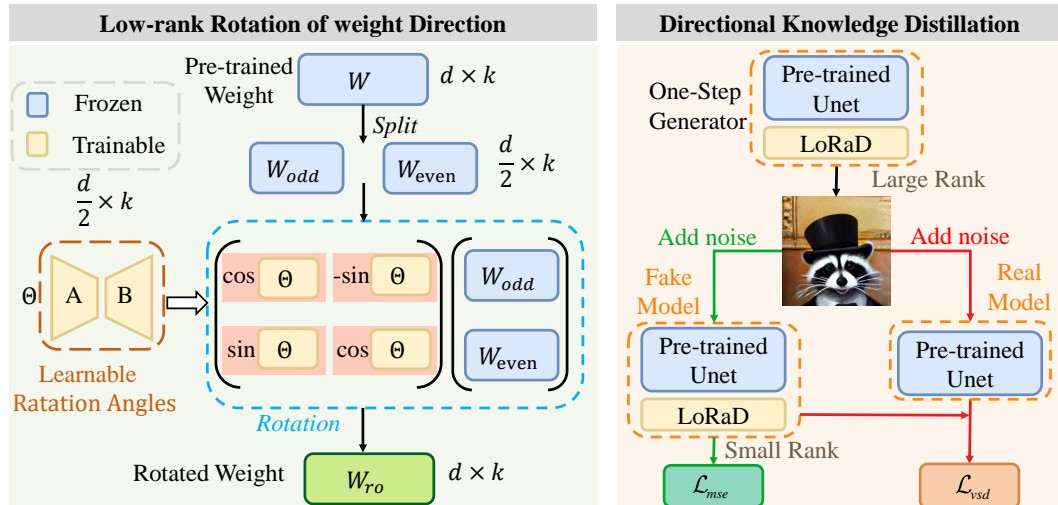

Figure 3: (Left) Detailed architecture of the Low-rank Rotation of weight Direction (LoRaD) module. The LoRaD rotates the pre-trained weight directions using learnable low-rank rotation angles. (Right) Overview of the Directional Knowledge Distillation (DKD) framework.

multi-step diffusion into a GAN. VSD adopts a dual-teacher strategy for distribution alignment (Dao et al., 2024; Nguyen & Tran, 2024; Zhou et al., 2024a; Yin et al., 2024a;b). SwiftBrush (Nguyen & Tran, 2024) achieves one-step, image-free generation. SwiftBrushv2 (Dao et al., 2024) leverages model ensembling, while DMD (Yin et al., 2024b) employs a regression loss to further improve performance. DMD2 (Yin et al., 2024a) extends VSD to few-step generation and underpins recent text-to-video acceleration frameworks (Yi et al., 2025; Shao et al., 2025).

However, existing training-based methods commonly use FT or LoRA, which can raise optimization difficulty. We find that directional changes are generally more influential in distillation. Therefore, we propose DKD, which leverages LoRaD to focus on modeling directional rotations.

## 3 METHOD

We first provide a brief overview of Variational Score Distillation (VSD) in Section 3.1, which serves as the foundation of our work. Motivated by the observation that weight direction changes play a key role in distillation, we introduce a *Low-rank Rotation of weight Direction* (LoRaD) module in Section 3.2 (See Appendix D for more theoretical explanation.). Finally, we integrate LoRaD into the VSD to form our proposed distillation framework, *Directional Knowledge Distillation* (DKD).

### 3.1 PRELIMINARY

**Latent Diffusion Models** (LDM) (Rombach et al., 2022) perform the diffusion process in a low-dimensional latent space, which improves computational efficiency. The training objective of LDM can be formulated as:

$$\mathcal{L}_{mse} = \min_{\varphi} \mathbb{E}_{t,\epsilon,\boldsymbol{c}} \left\| \epsilon_{\varphi} \left( \boldsymbol{z}_t, \boldsymbol{c}, t \right) - \epsilon \right\|_2^2, \tag{1}$$

where $\epsilon \sim \mathcal{N}(0, I)$ is Gaussian noise, $\boldsymbol{z}_t$ is the latent variable at timestep $t$, and $\boldsymbol{c}$ denotes the condition (*e.g.*, prompt) used to guide image generation. $\epsilon_{\varphi} \left( \boldsymbol{z}_t, \boldsymbol{c}, t \right)$ is the noise predicted by the model parameterized by $\varphi$.

**Variational Score Distillation** (VSD) (Wang et al., 2023) was initially proposed for text-to-3D generation to address issues such as oversaturation and reduced diversity. It was subsequently extended to 2D text-to-image generation in methods such as Swiftbrush (Nguyen & Tran, 2024), DMD (Yin et al., 2024b;a), and SiD (Zhou et al., 2024b;a), enabling one-step generation. The training objective of VSD is formulated as:

$$\nabla_{\lambda} \mathcal{L}_{vsd} = \mathbb{E}_{t,\epsilon,\boldsymbol{c}} \left[ \omega(t) \left( \epsilon_{\psi} \left( \boldsymbol{z}_t, \boldsymbol{c}, t \right) - \epsilon_{\phi} \left( \boldsymbol{z}_t, \boldsymbol{c}, t \right) \right) \frac{\partial G_{\lambda}(\boldsymbol{z}_{init}, \boldsymbol{c})}{\partial \lambda} \right], \tag{2}$$

where $\omega(t)$ is a time-dependent weighting term, $\epsilon_\psi$ is the real model parameterized by $\psi$, $\epsilon_\phi$ is the fake model parameterized by $\phi$, and $G_\lambda$ is the one-step generator parameterized by $\lambda$, with $z_{init} \sim \mathcal{N}(0, I)$ as its input noise. Additionally, $\epsilon_\phi$ is trained using Eq. (1). VSD alternates between updating $\epsilon_\phi$ and $G_\lambda$ until convergence.

## 3.2 Low-rank Rotation of Weight Direction

Analyzing the weight changes between multi-step U-Net models and their one-step counterparts suggests notable directional shifts with relatively small changes in norm. Motivated by this, we propose Low-rank Rotation of weight Direction (LoRaD) (see Fig. 3 (left)), which updates weights by learning rotations that alter only their directions. Furthermore, we observe that the changes in weight direction exhibit a low-rank structure (see Fig. 2 (b)). To exploit this property and reduce the overhead of full-rank modeling, which introduces additional parameters equivalent to 50% of the original weights, we adopt the low-rank decomposition strategy of LoRA (Hu et al., 2022). Starting from the 2D case ($d = 2$), given a weight vector $\alpha \in \mathbb{R}^d$, we apply a 2D rotation matrix as follows:

$$\alpha_{ro} = \begin{pmatrix} \cos\theta & -\sin\theta \\ \sin\theta & \cos\theta \end{pmatrix} \begin{pmatrix} \alpha^{(1)} \\ \alpha^{(2)} \end{pmatrix}, \tag{3}$$

where $\alpha_{ro}$ is the rotated weight vector. Inspired by the Rotary Position Embedding (RoPE) (Su et al., 2024), which generalizes the 2D case to any even dimension $d$, we apply a different rotation matrix[1] to each column of the pre-trained weight matrix $W \in \mathbb{R}^{d \times k}$:

$$W_{ro} = \left[ R_{\Theta,1}^d W_{\cdot,1}, R_{\Theta,2}^d W_{\cdot,2}, \cdots, R_{\Theta,k}^d W_{\cdot,k} \right], \tag{4}$$

where the rotation matrices $R_\Theta = \{R_{\Theta,i}^d\}_{i=1}^k$ are defined as:

$$R_{\Theta,i}^d = \begin{pmatrix} \cos\theta_{1,i} & -\sin\theta_{1,i} & 0 & 0 & \dots & 0 & 0 \\ \sin\theta_{1,i} & \cos\theta_{1,i} & 0 & 0 & \dots & 0 & 0 \\ 0 & 0 & \cos\theta_{2,i} & -\sin\theta_{2,i} & \dots & 0 & 0 \\ 0 & 0 & \sin\theta_{2,i} & \cos\theta_{2,i} & \dots & 0 & 0 \\ \vdots & \vdots & \vdots & \vdots & \ddots & \vdots & \vdots \\ 0 & 0 & 0 & 0 & \dots & \cos\theta_{\frac{d}{2},i} & -\sin\theta_{\frac{d}{2},i} \\ 0 & 0 & 0 & 0 & \dots & \sin\theta_{\frac{d}{2},i} & \cos\theta_{\frac{d}{2},i} \end{pmatrix}, \tag{5}$$

where $\Theta = \{\theta_j\}_{j=1}^{\frac{d}{2}} \in \mathbb{R}^{\frac{d}{2} \times k}$.

Given the sparsity of $R_{\Theta,i}^d$ in Eq. (5), the matrix-vector multiplication $R_{\Theta,i}^d W_{\cdot,i} \in \mathbb{R}^d$ can be computed efficiently as:

$$R_{\Theta,i}^d W_{\cdot,i} = \begin{pmatrix} W_{\cdot,i}^{(1)} \\ W_{\cdot,i}^{(2)} \\ W_{\cdot,i}^{(3)} \\ W_{\cdot,i}^{(4)} \\ \vdots \\ W_{\cdot,i}^{(d-1)} \\ W_{\cdot,i}^{(d)} \end{pmatrix} \odot \begin{pmatrix} \cos\theta_{1,i} \\ \cos\theta_{1,i} \\ \cos\theta_{2,i} \\ \cos\theta_{2,i} \\ \vdots \\ \cos\theta_{\frac{d}{2},i} \\ \cos\theta_{\frac{d}{2},i} \end{pmatrix} + \begin{pmatrix} W_{\cdot,i}^{(1)} \\ W_{\cdot,i}^{(2)} \\ W_{\cdot,i}^{(3)} \\ W_{\cdot,i}^{(4)} \\ \vdots \\ W_{\cdot,i}^{(d-1)} \\ W_{\cdot,i}^{(d)} \end{pmatrix} \odot \begin{pmatrix} -\sin\theta_{1,i} \\ \sin\theta_{1,i} \\ -\sin\theta_{2,i} \\ \sin\theta_{2,i} \\ \vdots \\ -\sin\theta_{\frac{d}{2},i} \\ \sin\theta_{\frac{d}{2},i} \end{pmatrix}, \tag{6}$$

where $\odot$ denotes element-wise multiplication. This implementation leverages the sparsity of the rotation matrix, allowing the computation to be performed using only element-wise operations, thus significantly reducing the computational cost.

Furthermore, since the rotation matrices in Eqs. (5) and (6) are block-diagonal with independent $2 \times 2$ submatrices, the computation can be efficiently implemented as a parallel application of multiple $2 \times 2$ rotations across odd-even index pairs. As shown in Fig. 3 (left), we split the $d$-dimensional space of the pre-trained weight matrix $W \in \mathbb{R}^{d \times k}$ into $\frac{d}{2}$ subspaces and rotate each independently. By separating the odd and even rows of $W$, we define:

$$W_{\text{odd}} = \left( W^{(1)}, W^{(3)}, \dots, W^{(d-1)} \right)^T,$$
$$W_{\text{even}} = \left( W^{(2)}, W^{(4)}, \dots, W_{\cdot,i}^{(d)} \right)^T, \tag{7}$$

---

[1]We do not need to explicitly separate the norm matrix, as rotations do not affect norm.

resulting in two matrices $W_{\text{odd}} \in \mathbb{R}^{\frac{d}{2} \times k}$ and $W_{\text{even}} \in \mathbb{R}^{\frac{d}{2} \times k}$.

The resulting parallel $2 \times 2$ rotations over each odd-even row pair can be expressed compactly as:

$$W_{ro} = R_\Theta W = \left[ \begin{array}{cc} \cos\Theta & -\sin\Theta \\ \sin\Theta & \cos\Theta \end{array} \right] \left[ \begin{array}{c} W_{\text{odd}} \\ W_{\text{even}} \end{array} \right], \tag{8}$$

where $W_{ro} \in \mathbb{R}^{d \times k}$ is the rotated weight matrix, and $\Theta \in \mathbb{R}^{\frac{d}{2} \times k}$ is the learnable rotation angle parameter matrix. To further reduce the number of trainable parameters, we apply low-rank decomposition to $\Theta$, inspired by LoRA (Hu et al., 2022), as follows:

$$\Theta = AB, \tag{9}$$

where $A \in \mathbb{R}^{\frac{d}{2} \times r}$ and $B \in \mathbb{R}^{r \times k}$ are low-rank parameter matrices with rank $r$. Finally, Eq. (8) can be rewritten as:

$$W_{ro} = R_\Theta W = R_{AB} W = \left[ \begin{array}{cc} \cos AB & -\sin AB \\ \sin AB & \cos AB \end{array} \right] \left[ \begin{array}{c} W_{\text{odd}} \\ W_{\text{even}} \end{array} \right]. \tag{10}$$

### 3.3 DIRECTIONAL KNOWLEDGE DISTILLATION

To fully leverage the directional characteristics observed in distillation, we integrate LoRaD into the VSD. This yields a direction-aware distillation framework, which we term Directional Knowledge Distillation (DKD). As illustrated in Fig. 3 (right), DKD employs a pre-trained diffusion model $\epsilon_\psi$ as the teacher (real model) and introduces a trainable fake model $\epsilon_\phi$ (initialized from $\epsilon_\psi$) to approximate the teacher's distribution. The final student model (one-step generator) $G_\lambda$, also initialized from $\epsilon_\psi$, is trained to synthesize high-quality images in one-step. See Appendix F.3 for algorithm details.

To enhance alignment with the real distribution, we apply LoRaD to both the student and fake models. Specifically, the one-step generator $G_{\lambda_{\Theta^l}}$ incorporates a high-rank rotation matrix $\Theta^l$ to better fit the teacher, while the fake model $\epsilon_{\phi_{\Theta^s}}$ uses a low-rank rotation matrix $\Theta^s$ to provide adaptive guidance. Finally, we alternate the optimization of $\lambda_{\Theta^l}$ and $\phi_{\Theta^s}$ to jointly improve the quality of the generation.

Accordingly, the DKD training objective can be rewritten from Eq. (2) as:

$$\nabla_{\lambda_{\Theta^l}} \mathcal{L}_{vsd} = \mathbb{E}_{t,\epsilon,\boldsymbol{c}} \left[ \omega(t) \left( \epsilon_\psi \left( \boldsymbol{z}_t, \boldsymbol{c}, t \right) - \epsilon_{\phi_{\Theta^s}} \left( \boldsymbol{z}_t, \boldsymbol{c}, t \right) \right) \frac{\partial G_{\lambda_{\Theta^l}} \left( \boldsymbol{z}_{init}, \boldsymbol{c} \right)}{\partial \lambda_{\Theta^l}} \right], \tag{11}$$

The training objective for $\epsilon_{\phi_{\Theta^s}}$ can also be rewritten from Eq. (1) as:

$$\min_{\phi_{\Theta^s}} \mathbb{E}_{t,\epsilon,\boldsymbol{c}} \left\| \epsilon_{\phi_{\Theta^s}} \left( \boldsymbol{z}_t, \boldsymbol{c}, t \right) - \epsilon \right\|_2^2. \tag{12}$$

## 4 EXPERIMENT

### 4.1 EXPERIMENTAL SETUP

**Evaluation Datasets and Metrics.** We systematically evaluate the zero-shot text-to-image generation capability of DKD on the COCO 2014 (Lin et al., 2014) and COCO 2017 (Lin et al., 2014) datasets, using 30k and 5k randomly sampled images, respectively. To comprehensively assess the quality of the generation, we use the Fréchet Inception Distance (FID) (Heusel et al., 2017) to measure image fidelity and the CLIP score (Radford et al., 2021) to evaluate the semantic alignment of text-image. The FID is calculated using Inception V3 (Szegedy et al., 2016) as the feature extractor, while the CLIP score is based on the ViT-G/14 (Cherti et al., 2023) model. We further adopt precision and recall (Kynkäänniemi et al., 2019) to evaluate fidelity and diversity. Finally, we also evaluate text-image alignment on the Human Preference Score v2 (HPSv2) (Wu et al., 2023b) benchmark. See Appendix G.1 for details.

**Implementation Details.** Following prior methods (Nguyen & Tran, 2024; Dao et al., 2024; Yin et al., 2024a;b), the student model in DKD adopts the same architecture as the teacher and is initialized with the teacher's weights. DKD is trained on 1.4 M prompts sampled from the JourneyDB (Sun et al., 2023) dataset. During training, the learning rate (LR) for the student is set to $1e$-4, while the

Table 1: Quantitative comparison of DKD and other methods on zero-shot COCO 2014 results. $^*$ indicates our reproduced results, and $^\wr$ indicates results using the official pre-trained models. '-' denotes unknown. Best and second-best scores are in **bold** and underline, respectively. Image-free" refers to training without supervision from real images.

| Method | #Params | NFEs | Type | Trainable params | FID↓ | CLIP↑ | Precision↑ | Recall↑ | Image-free? | Training Data |
|---|---|---|---|---|---|---|---|---|---|---|
| Stable Diffusion 1.5-based backbone | | | | | | | | | | |
| SD 1.5 ($cfg = 3.0$) | 860M | 25 | U-Net | 860M | 8.78 | 0.30 | 0.59 | 0.53 | ✗ | 5B |
| LCM-LoRA$^\wr$ | 860M | 1 | LoRA | 67.50M | 77.73 | 0.24 | 0.22 | 0.15 | ✗ | 12M |
| InstaFlow | 860M | 1 | U-Net | 860M | 13.10 | 0.28 | 0.53 | 0.45 | ✗ | 3.2M |
| UFOGen | 860M | 1 | U-Net | 860M | 12.78 | - | - | - | ✗ | 12M |
| DMD | 860M | 1 | U-Net | 860M | 11.49 | **0.32** | - | - | ✗ | 3M |
| DMD2* | 860M | 1 | U-Net | 860M | 12.96 | 0.30 | 0.60 | 0.47 | ✓ | 1.4M |
| SiD-LSG* | 860M | 1 | U-Net | 860M | 14.27 | 0.30 | 0.56 | **0.48** | ✓ | 1.4M |
| PCM | 860M | 1 | U-Net | 860M | 17.91 | 0.29 | - | - | ✗ | 3M |
| Hyper-SD$^\wr$ | 860M | 1 | LoRA | 67.25M | 22.90 | 0.31 | **0.62** | 0.25 | ✗ | - |
| YOSO$^\wr$ | 860M | 1 | LoRA | 67.25M | 23.68 | 0.29 | 0.56 | 0.36 | ✗ | 4M |
| DKD | 860M | 1 | LoRaD | 83.80M | **10.79** | 0.31 | **0.62** | **0.48** | ✓ | 1.4M |
| Stable Diffusion 2.1-based backbone | | | | | | | | | | |
| SD 2.1 ($cfg = 3.0$) | 865M | 1 | U-Net | 865M | 9.60 | 0.32 | 0.59 | 0.50 | ✗ | 5B |
| SD-Turbo$^\wr$ | 865M | 1 | U-Net | 865M | 16.14 | **0.33** | **0.65** | 0.35 | ✗ | - |
| Swiftbrush | 865M | 1 | U-Net | 865M | 16.67 | 0.29 | 0.47 | 0.46 | ✓ | 1.4M |
| Swiftbrushv2* | 865M | 1 | U-Net+LoRA | 884.14M | 15.98 | **0.33** | 0.58 | 0.47 | ✓ | 1.4M |
| SiD-LSG* | 865M | 1 | U-Net | 865M | 15.17 | 0.30 | 0.56 | 0.46 | ✓ | 1.4M |
| TiUE$^\wr$ | 865M | 1 | U-Net | 865M | 13.49 | 0.31 | 0.59 | **0.48** | ✓ | 1.4M |
| DKD | 865M | 1 | LoRaD | 94.43M | **12.34** | 0.31 | 0.60 | **0.48** | ✓ | 1.4M |
| PixArt-$\alpha$-based backbone | | | | | | | | | | |
| PixArt-$\alpha$ ($cfg = 4.5$)$^\wr$ | 610.86M | 20 | DiT | 610.86M | 8.75 | 0.32 | 0.75 | 0.45 | ✗ | 25M |
| Swiftbrush* | 610.86M | 1 | DiT | 610.86M | 29.89 | 0.28 | 0.50 | 0.26 | ✓ | 1.4M |
| PG-SB* | 610.86M | 1 | DiT | 610.86M | 25.58 | 0.28 | 0.53 | 0.27 | ✓ | 1.4M |
| DKD | 610.86M | 1 | LoRaD | 81.22M | **18.99** | **0.30** | **0.64** | **0.29** | ✓ | 1.4M |

fake model uses a LR of $1e$-2. We use AdamW (Loshchilov & Hutter, 2019) as the optimizer, with a batch size of 128 (16 per GPU). The classifier-free guidance (CFG) scale is set to 1.5, and the training is conducted for 2 epochs. We distill student models based on three different backbones, namely SD 1.5 (Rombach et al., 2022), SD 2.1 (Rombach et al., 2022), and PixArt-$\alpha$ ($256 \times 256$) (Chen et al., 2023). For SD 1.5 and SD 2.1, the LoRaD rank of the student is set to 256, while for PixArt-$\alpha$, it is set to 128. The LoRaD rank for all fake models is uniformly set to 32. See Appendix F.1 for details.

## 4.2 Comparison with State-of-the-Art Methods

**Quantitative results.** We comprehensively evaluate DKD on the COCO 2014 dataset against SOTA zero-shot one-step generation methods across three backbones: SD 1.5, SD 2.1, and PixArt-$\alpha$. To ensure fair comparison and considering computational constraints, we follow the setup of TiUE (Li et al., 2025) and uniformly reproduce DKD, DMD2, SiD-LSG, and SwiftBrushv2 using 1.4M prompts. As shown in Tab. 1, DKD achieves the best FID and Recall scores on all backbones, demonstrating superior fidelity and diversity. It also ranks first or second in CLIP and Precision, indicating strong text-image alignment and perceptual quality. Notably, only 9.74%, 10.92%, and 13.30% of the model parameters are trainable for SD 1.5, SD 2.1, and PixArt-$\alpha$, respectively, highlighting DKD's parameter efficiency. These improvements stem from our proposed LoRaD, which reparameterizes weight updates via low-rank rotations to enable stable and efficient distillation. See Appendix F.4, G.3.

**Qualitative results.** Fig. 4 presents a qualitative comparison of DKD with SOTA one-step generation methods based on SD 1.5 and SD 2.1 backbones. Across diverse prompts, DKD consistently produces visually coherent and semantically aligned results. For example, in the first and second rows, DKD better preserves structure and stylistic fidelity, capturing sharp features and vibrant colors without artifacts or distortions. In the third and fourth rows, it accurately follows prompts involving specific subjects (*e.g.*, sphynx cat, corgi, shiba inu) and contexts (*e.g.*, theater, clothing), while alternative methods often miss key attributes or yield unrealistic shapes. Notably, in the last row, DKD generates complex scenes (*e.g.*, dog looking at TV) with consistent spatial composition and background details, demonstrating superior holistic understanding compared to other baselines. See Appendix G.5.

## 4.3 Downstream Tasks

**Controllable generation.** ControlNet (Zhang et al., 2023) is a widely used controllable generation model that incorporates spatial conditions into SD (Rombach et al., 2022) for fine-grained control. As shown in Fig. 5, applying DKD to ControlNet significantly improves inference efficiency, reducing inference time by **86.26%** while preserving image quality, faithfully following spatial conditions, and maintaining prompt adherence comparable to ControlNet.

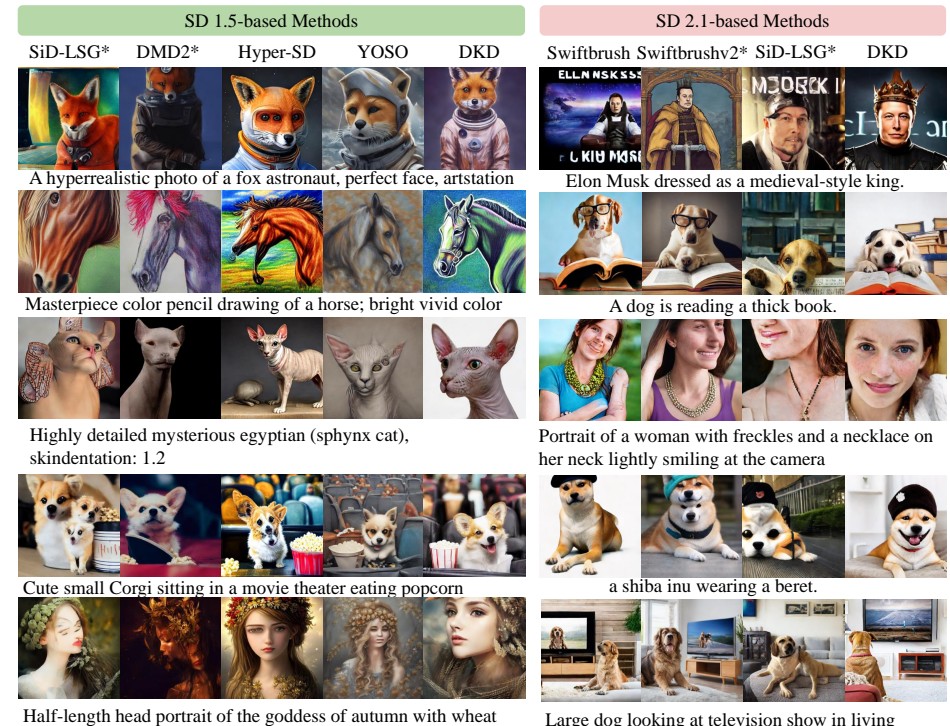

Figure 4: Qualitative comparison with other methods, where * indicates our reproduced results.

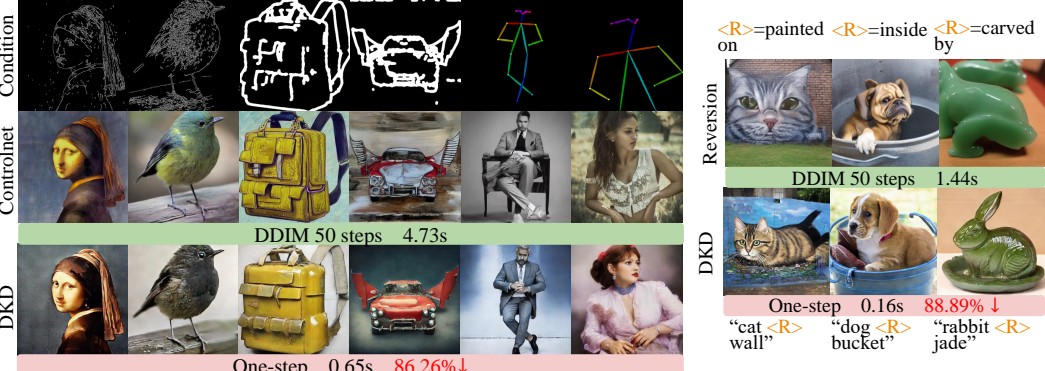

Figure 5: Quality results by Controlnet (Zhang et al., 2023) with or without DKD.

Figure 6: Quality results by Reversion (Huang et al., 2024b) with or without DKD.

**Relation inversion.** Reversion (Huang et al., 2024b) is the first method to guide specific object relationship synthesis in SD via relational prompts. Integrating DKD into Reversion significantly accelerates inference. As shown in Fig. 6, DKD reduces inference time by **88.89%** while producing high-fidelity images that align with the relational prompts, with quality close to that of the original multi-step Reversion. See Appendix F.2 for more results.

Table 2: Ablation study on the impact of rank in DKD (SD 1.5) on the COCO 2017 dataset. "N" and "DM" denote the norm mean and direction mean for all layers, respectively.

| Type | #Params | FID | CLIP | NM | DM |
|------|---------|-----|------|-----|-----|
| LoRA | 120.9M | 25.27 | 0.29 | 0.06 | 0.83 |
| DoRA | 121.2M | 26.56 | 0.30 | 0.03 | 0.55 |
| FT (DMD2) | 860.0M | 23.30 | 0.30 | 0.10 | 2.21 |
| LoRaD | **83.8M** | **20.86** | **0.31** | - | 2.89 |

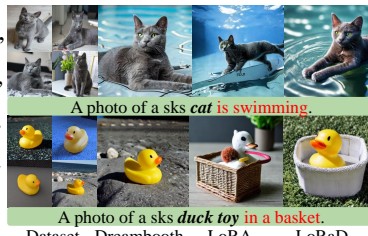

Figure 7: Quality results by Dreambooth with or without LoRaD.

Table 3: Ablation study on the impact of the rank on DKD (SD 1.5) on COCO 2014 dataset.

| Setting | Rank | | | | FID | CLIP |
|---------|------|--------|------|--------|-----|------|
| | Student | #Params | Fake model | #Params | | |
| A | 64 | 20.95M | 32 | 9.38M | 13.64 | 0.30 |
| B | 128 | 41.90M | 32 | 9.38M | 13.16 | 0.29 |
| C | 256 | 83.80M | 32 | 9.38M | **10.79** | **0.31** |
| D | 512 | 167.59M | 32 | 9.38M | 12.75 | 0.30 |
| E | 256 | 83.80M | 16 | 4.69M | 17.53 | 0.29 |
| F | 256 | 83.80M | 64 | 18.76M | 16.98 | **0.31** |

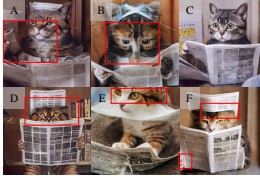

Figure 8: One-step image generation with various settings.

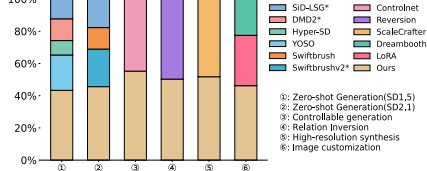

Figure 9: User study results compared to other methods.

**Image customization.** Dreambooth (Ruiz et al., 2023) is a pioneering personalized text-to-image framework that binds the target subject to a rare token via FT of the U-Net. To enhance parameter efficiency, we integrate our proposed LoRaD into Dreambooth and compare it with Dreambooth (FT) and LoRA (Hu et al., 2022). As shown in Fig. 7, vanilla DreamBooth overfits by capturing the subject while memorizing training images, thus reducing prompt sensitivity. LoRA alleviates overfitting, but degrades subject identity and image fidelity. In contrast, LoRaD maintains subject fidelity while adhering to prompts, achieving a better balance. These results highlight the generalizability of LoRaD beyond distillation, motivating future applications in broader vision tasks requiring fine-tuning.

### 4.4 USER STUDY

To evaluate image quality and text-image alignment, we conducted a user study with 57 participants, covering zero-shot generation and downstream tasks. As shown in Fig. 9, the results clearly demonstrate the superiority of our method over existing baselines. See Appendix F.5 for details.

### 4.5 ABLATION STUDIES

Tab. 2 compares the performance of four different fine-tuning types (LoRA, DoRA, FT, and LoRaD) on the COCO 2017 dataset. LoRaD attains the best scores (FID 20.86, CLIP 0.31) with the fewest trainable parameters (83.8M; 31% fewer than LoRA/DoRA and 90% fewer than FT). Moreover, LoRaD achieves the highest direction mean (2.89% vs. 2.21% for FT, 0.83%/0.55% for LoRA/DoRA), suggesting a broader and more effective update direction space under a compact parameterization. Overall, the results indicate a favorable quality–efficiency trade-off for LoRaD.

We conduct an ablation study on the COCO 2014 dataset to assess the impact of rank configuration in DKD. As shown in Tab. 3, we make three key observations: 1) *Increasing student rank consistently improves performance.* Raising the rank from setting A to C reduces FID from 13.64 to 10.79, indicating that higher rank enables the student to better capture the teacher's distribution and improve generation quality. 2) *Increasing the rank beyond a threshold yields diminishing returns.* Comparing settings C and D, further increasing the rank degrades FID (12.75 vs. 10.79) and CLIP (0.31 vs. 0.30), suggesting that overly large ranks may cause overfitting. 3) *Fake model rank affects fidelity more than alignment.* Varying the fake model rank (settings C, E, F) changes FID but leaves CLIP largely stable, implying fidelity is more sensitive to capacity than alignment. In summary, setting C offers a favorable trade-off between model capacity and performance, consistent with the qualitative results in Fig. 8. See Appendix G.2, G.4 for details.

### 5 CONCLUSION

This paper presents Directional Knowledge Distillation (DKD), an efficient one-step text-to-image distillation framework. Through an in-depth analysis of weight changes between multi-step and one-step models, we find that changes in weight direction serve as a key mechanism in distillation, while changes in norm play a comparatively smaller role. Based on this insight, we introduce the Low-rank Rotation of weight Direction (LoRaD) module to model directional adjustments in a parameter-efficient manner. Extensive experiments demonstrate that DKD significantly outperforms existing one-step methods—such as DMD, SiD-LSG, and SwiftBrush—in both image quality and inference speed. Moreover, the distilled model can be seamlessly adapted to a wide range of downstream tasks, showcasing strong generalization and practical applicability. Our work offers a novel theoretical perspective and practical solution for efficient diffusion model distillation.

## ETHICS STATEMENT

We fully recognize the potential ethical risks associated with deploying generative models, including privacy breaches, data misuse, and the amplification of biases. At the same time, we acknowledge the potential misuse of personalization and customization techniques in generating false content and disinformation. To address these concerns, we advocate for and support responsible research and application practices, strictly adhering to relevant laws, regulations, and industry guidelines, while implementing necessary technical and governance measures to minimize the risks of misuse.

## REPRODUCIBILITY STATEMENT

To promote reproducibility, we will release all source code and scripts after the peer review process, enabling others to replicate the experiments. All experiments in this work were conducted using publicly available datasets.

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

Table 4: Comparison of different methods in terms of memory, number of trainable parameters, FID, CLIP score, and latency.

| Type | Memory (M) | #Train Param. | FID | CLIP | Latency |
|---|---|---|---|---|---|
| LoRA | 1259 | 120.9M | 25.27 | 0.29 | 0.11s |
| LoRaD | 2021 | **83.8M** | **20.86** | **0.31** | 0.11s |
| FT (DMD2) | 17397 | 860M | 23.30 | 0.30 | 0.11s |

## A  LIMITATIONS AND FUTURE WORK

Assuming the weight matrix $W \in \mathbb{R}^{d \times k}$, the training memory usage of full fine-tuning (FT) is $2 \times d \times k$, while that of LoRA is $2 \times (d \times r + r \times k)$. In comparison, LoRaD requires $d \times k + 2 \times \left(\frac{d}{2} \times r + r \times k\right) = d \times k + d \times r + 2 \times r \times k$, reflecting higher memory overhead than LoRA due to the need to reconstruct a full-rank rotated weight matrix during forward pass. Note that this operation is performed only in the forward pass and does not incur additional gradient storage. As shown in Tab. 4, LoRA uses ∼7% of DMD2's memory, while LoRaD uses slightly more at ∼12%. Future work may explore more memory-efficient rotation strategies that avoid explicit construction of the full rotated matrix. Additionally, given the flexibility and generality LoRaD has demonstrated in the image customization task, we plan to further investigate its applicability to a broader range of model fine-tuning tasks.

## B  LLM USAGE STATEMENT

We only employed a large language model (ChatGPT) for polishing the language and improving the clarity of expression. The model was not involved in the conception or design of the research, the development or execution of methods or experiments, data processing or statistical analysis, nor in the interpretation of results or the derivation of conclusions; therefore, it had no substantive impact on the core scientific contributions of this work.

## C  BROADER IMPACTS

DKD compresses the inference process of diffusion models into one-step through knowledge distillation, significantly reducing computational resource consumption. At the same time, it promotes creativity among content generators and lowers the barrier to entry. However, the technology may also be misused to generate false or harmful images, thereby spreading misinformation and raising concerns related to copyright and intellectual property.

## D  MORE THEORETICAL EXPLANATION

While our empirical results demonstrate that knowledge distillation primarily preserves weight directions, we further provide a theoretical perspective to support this observation. When optimizing only the directional component of weights, the parameters are constrained to lie on a unit hypersphere—a compact and smooth Riemannian manifold (Absil et al., 2008; Boumal, 2023). According to manifold optimization theory (Liu & Zhu, 2018; Han et al., 2021), such geometric constraints improve gradient flow and mitigate sharp or pathological minima, thereby stabilizing training.

In addition, constraining optimization to the hypersphere reduces the degrees of freedom in parameter space, serving as an implicit regularizer. This aligns with modern generalization theory in overparameterized settings, which suggests that limiting parameter complexity improves robustness (Beik-Mohammadi et al., 2023; Barp et al., 2022). LoRaD explicitly models directional changes through learnable 2D rotations, analogous to reparameterizations used in weight normalization (Salimans & Kingma, 2016), which have been shown to accelerate convergence and enhance generalization (Salimans & Kingma, 2016; Liu et al., 2024).

Optimizing directions alone substantially reduces sensitivity to weight norm and tends to produce smoother loss landscapes (Lyu et al., 2022), favoring flatter minima—empirically associated with

better generalization (Kaddour et al., 2022; Arora et al., 2018). These effects are particularly important for robustness under distribution shifts (Fei et al., 2025). This theoretical grounding is consistent with our empirical findings, including the superior performance of LoRaD shown in Tab. 1 and its convergence behavior in Fig. 16 and Fig. 17.

## E   MORE DETAILS ON WEIGHT ANALYSIS

In our weight analysis, we decompose the weight matrix $W \in \mathbb{R}^{d \times k}$ into a norm vector and direction matrix as follows:

$$W = \eta \mathcal{V} \tag{13}$$

where $\eta \in \mathbb{R}^{1 \times k}$ represents the column-wise norm, and $\mathcal{V} \in \mathbb{R}^{d \times k}$ denotes the normalized direction matrix.

To quantify the difference between the multi-step and one-step U-Net weights, we compute the mean and standard deviation (STD) of the changes in their norm and direction:

- Changes in the mean norm:

$$\Delta \eta_{\text{mean}} = \frac{1}{k} \sum_{i=1}^{k} \frac{\left| \eta_{\text{one-step}}^i - \eta_{\text{multi-step}}^i \right|}{\eta_{\text{multi-step}}^i} \tag{14}$$

- Changes in the mean direction:

$$\Delta \mathcal{V}_{\text{mean}} = \frac{1}{k} \sum_{i=1}^{k} \sqrt{\sum_{j=1}^{d} \left( \mathcal{V}_{\text{one-step}}^{i,j} - \mathcal{V}_{\text{multi-step}}^{i,j} \right)^2}. \tag{15}$$

- Changes in the STD norm:

$$\Delta \eta_{\text{std}} = \sqrt{\frac{1}{k} \sum_{i=1}^{k} \left( \frac{\left| \eta_{\text{one-step}}^i - \eta_{\text{multi-step}}^i \right|}{\eta_{\text{multi-step}}^i} - \Delta \eta_{\text{mean}} \right)^2} \tag{16}$$

- Changes in the STD direction:

$$\Delta \mathcal{V}_{\text{std}} = \sqrt{\frac{1}{k} \left( \sum_{i=1}^{k} \sqrt{\sum_{j=1}^{d} \left( \mathcal{V}_{\text{one-step}}^{i,j} - \mathcal{V}_{\text{multi-step}}^{i,j} \right)^2} - \Delta \mathcal{V}_{\text{mean}} \right)^2} \tag{17}$$

As shown in Fig. 10, Fig. 11, Fig. 12, and Fig. 13, we also provide more visualization results including Swiftbrush (Nguyen & Tran, 2024), Hyper-SD (Ren et al., 2024), SD-Turbo (Sauer et al., 2024), YOSO (Luo et al., 2024), SiD-LSG (Zhou et al., 2024a), Swiftbrushv2 (Dao et al., 2024), and PCM (Wang et al., 2024), etc. with SD 1.5, SD 2.1 or Pixart-$\alpha$ as the backbone. Similar conclusions can be drawn as in Sec. 1.

## F   IMPLEMENTATION DETAILS

### F.1   TRAINING AND INFERENCE DETAILS

**Zero-shot generation.** 1) We implement DKD using PyTorch and optimize it with the AdamW (Loshchilov & Hutter, 2019) optimizer ($\beta_1 = 0.9$, $\beta_2 = 0.999$). LoRaD is applied to all linear layers in the U-Net/DiT architecture, including the feedforward layers, time_emb_proj layers, projection layers, and $Q, K, V$, out layers. Although we observe similar directional changes in convolutional layers, applying LoRaD to them introduces additional parameter overhead and lacks generality, as architectures like DiT (Chen et al., 2023) and autoregressive (Sun et al., 2024) models are primarily composed of linear layers. Notably, applying LoRaD solely to linear layers is sufficient to achieve SOTA performance. 2) We reproduce DMD2 (Yin et al., 2024a), SiD-LSG (Zhou et al.,

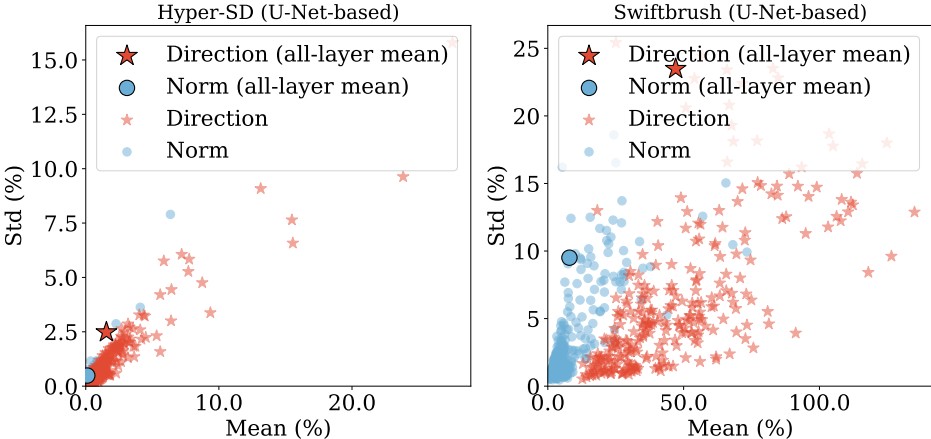

Figure 10: Visualization of changes in weight norm and direction (Hyper-SD (Ren et al., 2024) and Swiftbrush (Nguyen & Tran, 2024)).

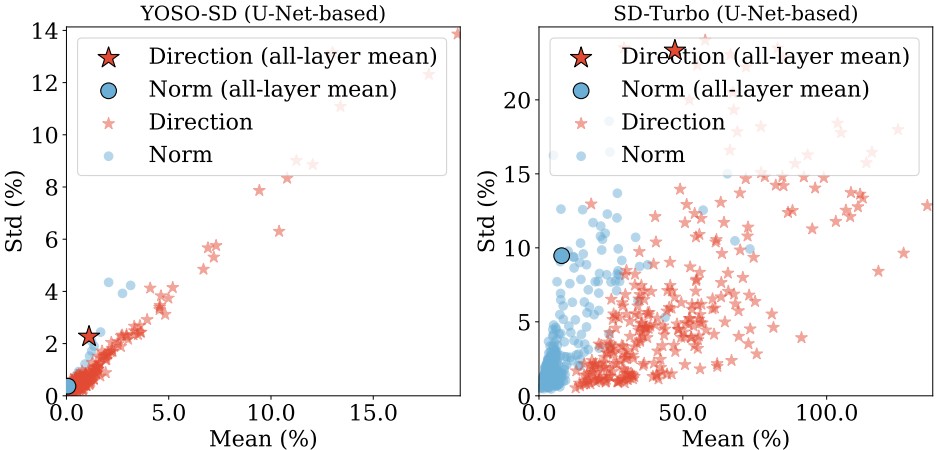

Figure 11: Visualization of changes in weight norm and direction (YOSO (Luo et al., 2024) and SD-Turbo (Sauer et al., 2024)).

2024a), SwiftBrush (Nguyen & Tran, 2024), and PG-SB (Nguyen et al., 2024), following the default hyperparameter settings reported in their respective papers, except for using JourneyDB (Sun et al., 2023) as the training dataset. Specifically, our reproduction of DMD2 adopts the image-free variant without the second-stage GAN loss. For SiD-LSG, the guidance scales $k_1, k_2, k_3, k_4$ are set to 1.5.

**Baselines.** We conduct DKD evaluations against a range of baselines on different model backbones. For the SD 1.5-based (Rombach et al., 2022) backbone, we select LCM-LoRA (Luo et al., 2023b), InstaFlow (Liu et al., 2023), UFOGen (Xu et al., 2024), DMD (Yin et al., 2024b), DMD2 (Yin et al., 2024a), SiD-LSG (Zhou et al., 2024a), PCM (Wang et al., 2024), Hyper-SD (Ren et al., 2024), and YOSO (Luo et al., 2024) as baselines. For the SD 2.1-based (Rombach et al., 2022) backbone, we compare with SD-Turbo (Sauer et al., 2024), SwiftBrush (Nguyen & Tran, 2024), SwiftBrushv2 (Dao et al., 2024), SiD-LSG (Zhou et al., 2024a), and TiUE (Li et al., 2025). For the DiT-based (Chen et al., 2023) backbone, SwiftBrush (Nguyen & Tran, 2024) and PG-SB (Nguyen et al., 2024) are chosen as baselines. In addition, we apply DKD to four downstream tasks, including controllable generation, relation inversion, high-resolution synthesis, and image customization. For these tasks, we use ControlNet (Zhang et al., 2023), Reversion (Huang et al., 2024b), ScaleCrafter (He et al., 2023), DreamBooth (Ruiz et al., 2023), and LoRA (Hu et al., 2022) as baselines.

**Downstream tasks.** For all tasks except image personalization, we replace the original multi-step U-Net with our distilled one-step DKD model to accelerate inference. For image customization,

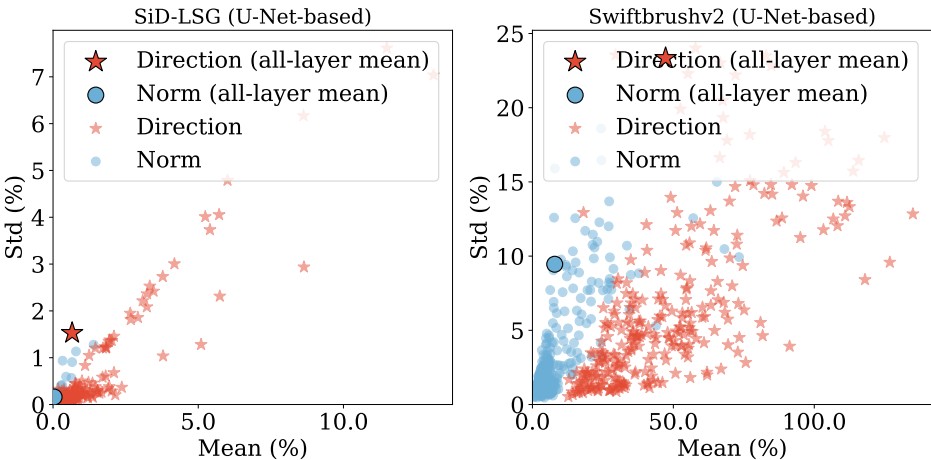

Figure 12: Visualization of changes in weight norm and direction (SiD-LSG (Zhou et al., 2024a) and Swiftbrushv2 (Dao et al., 2024)).

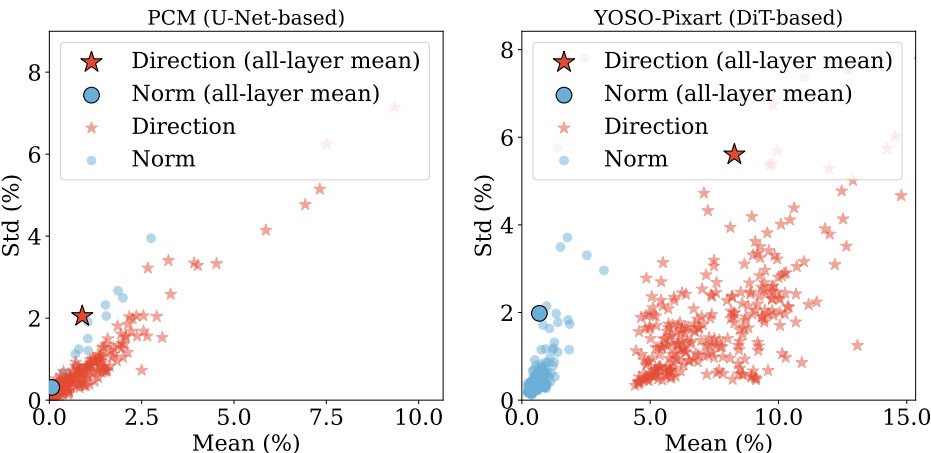

Figure 13: Visualization of changes in weight norm and direction (PCM (Wang et al., 2024) and YOSO-Pixart (Luo et al., 2024)).

we apply LoRaD during fine-tuning. 1) During inference with ControlNet (Zhang et al., 2023), Reversion (Huang et al., 2024b), ScaleCrafter (He et al., 2023), and DreamBooth (Ruiz et al., 2023), we use the DDIM (Song et al., 2021a) scheduler with 50 inference steps. 2) In implementing DreamBooth, LoRA, and LoRaD, we adopt the prior-preservation loss with a weight factor of 1.0, and use the Adam (Kingma & Ba, 2014) optimizer. The number of class-conditioned images is set to 200. For DreamBooth, the learning rate (LR) is set to 5e-6 with 800 training steps. For both LoRA and LoRaD, the LR is set to 1e-4, the number of training steps is 1500, and the rank is set to 64.

## F.2 MORE DESCRIPTIONS OF DOWNSTREAM TASKS.

Controllable generation introduces structured control signals (e.g., edge maps, human pose, or depth maps) into text-to-image models to guide the generation process toward producing images that not only match the textual description but also conform to specific spatial or semantic constraints. By incorporating external structural priors, controllable generation enables fine-grained manipulation of object layout, shape, and orientation—capabilities that are difficult to achieve with pure text prompts alone. Controllable generation plays an important role in tasks such as conditional image synthesis and creative visual design, where strict adherence to user-provided structure is crucial for usability and reliability.

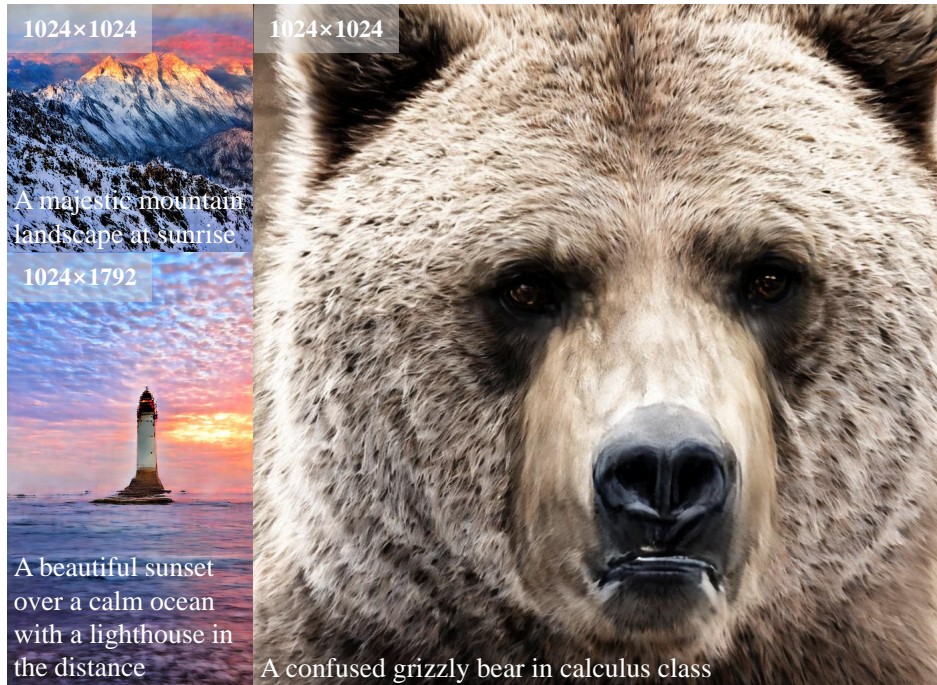

Figure 14: Quality results by ScaleCrafter (He et al., 2023) with DKD.

Relation inversion aims to learn specific relational patterns (e.g., handshake, back-to-back, hugging) from a few input images and encode them into relation prompts, which can then be generalized to novel subjects, poses, and styles. Instead of focusing on individual object identity or appearance, relation inversion captures the spatial and semantic relationships between multiple entities, allowing the model to reproduce similar interactions under new contexts. Relation inversion enables controllable multi-subject generation and relational transfer.

High-resolution synthesis aims to generate high-resolution images (e.g., 512×512 or higher) with both strong global coherence and rich local detail. Compared to low-resolution generation, this task presents greater challenges in terms of spatial consistency, object fidelity, and fine-grained texture reconstruction. High-resolution synthesis requires the model to capture long-range dependencies across the image while preserving subtle local variations such as edges, patterns, and shading. Despite these challenges, it plays a crucial role in applications such as photo-realistic content creation and artistic image generation.

Image customization aims to learn the visual concept of a specific subject from a few example images and generalize it to new images. Specifically, it involves fine-tuning a pre-trained model to capture and retain the subject's distinctive visual features, enabling the model to generate personalized images that preserve the subject identity when given new text prompts. This capability is particularly valuable in applications such as creative design, personalized content generation, and artistic image synthesis.

**High-resolution synthesis.** ScaleCrafter (He et al., 2023) enables variable-resolution image generation in SD without retraining, which is accomplished by adjusting receptive fields within the pre-trained U-Net. To address the fixed-resolution limitation inherited from SD 1.5/2.1, we integrate DKD with ScaleCrafter. As illustrated in Fig. 14, this combination allows DKD to generate high-fidelity images at varying resolutions, demonstrating strong scalability.

### F.3 Pseudo training code

As shown in Algorithm 1, we provide the pseudocode for DKD to clearly outline the key steps of the algorithm.

---

**Algorithm 1** DKD: Directional Knowledge Distillation

---

1: **Require:** Pretrained real model $\epsilon_\psi$, fake model $\epsilon_{\phi_{\Theta^s}}$, one-step generator $G_{\lambda_{\Theta^l}}$, learning rates $\gamma_1$ and $\gamma_2$, initial timestep $t_{\text{init}}$, time-dependent weight function $\omega(t)$, prompts dataset $\mathcal{D}$, maximum number of timesteps $T_{\max}$ and scheduler schedule $\{(\alpha_t, \sigma_t)\}_{t=1}^{t=T}$

2: **Initialize:** $\phi_{\Theta^s} \leftarrow \psi$, $\lambda_{\Theta^l} \leftarrow \psi$, $\gamma_1 = 1e-4$, $\gamma_2 = 1e-2$, $t_{\text{init}} = 999$

3: **repeat**

4:  Sample input noise $\boldsymbol{z} \sim \mathcal{N}(0, I)$ and prompt $\boldsymbol{c} \sim \mathcal{D}$

5:  Generate one-step output $\boldsymbol{x}_0 = G_{\lambda_{\Theta^l}}(\boldsymbol{z}, \boldsymbol{c}, t_{\text{init}})$

6:  Sample timestep $t \sim \mathcal{U}(0.02T_{\max}, 0.98T_{\max})$ and noise $\epsilon \sim \mathcal{N}(0, I)$

7:  Compute noisy latent code $\boldsymbol{x}_t = \alpha_t \boldsymbol{x}_0 + \sigma_t \epsilon$

8:  Update $G_{\lambda_{\Theta^l}}$ with $\lambda_{\Theta^l} \leftarrow \lambda_{\Theta^l} - \gamma_1 \left[ \omega(t) \left( \epsilon_\psi(\boldsymbol{x}_t, \boldsymbol{c}, t) - \epsilon_{\phi_{\Theta^s}}(\boldsymbol{x}_t, \boldsymbol{c}, t) \right) \frac{\partial \boldsymbol{x}_0}{\partial \lambda_{\Theta^l}} \right]$

9:  Sample another timestep $\tilde{t} \sim \mathcal{U}(0.02T_{\max}, 0.98T_{\max})$ and noise $\tilde{\epsilon} \sim \mathcal{N}(0, I)$

10:  Compute noisy latent code $\boldsymbol{x}_{\tilde{t}} = \alpha_{\tilde{t}} \boldsymbol{x}_0 + \sigma_{\tilde{t}} \tilde{\epsilon}$

11:  Update $\epsilon_{\phi_{\Theta^s}}$ with $\Theta^s \leftarrow \Theta^s - \gamma_2 \nabla_{\phi_{\Theta^s}} \left\| \epsilon_{\phi_{\Theta^s}}(\boldsymbol{x}_{\tilde{t}}, \tilde{t}, \boldsymbol{c}) - \tilde{\epsilon} \right\|^2$

12: **until** processing 1.4M prompts or training budget is exhausted

13: **return** Trained one-step generator $G_{\lambda_{\Theta^l}}$

---

## F.4 COMPARISON OF TRAINING AND INFERENCE TIME

Table 5: Comparison of inference and training times of our method vs. other methods on the zero-shot benchmark of COCO 2014. * indicates our reproduced results, and $^l$ indicates results using the official pre-trained models. '-' denotes unknown. Best and second-best scores are in **bold** and underline, respectively.

| Method | NFEs | Type | Trainable params | FID↓ | CLIP↑ | Image-free? | Inference | A100 Days |
|---|---|---|---|---|---|---|---|---|
| Stable Diffusion 1.5-based backbone | | | | | | | | |
| SD 1.5 (*cfg* = 3.0) (Rombach et al., 2022) | 25 | U-Net | 860M | 8.78 | 0.30 | ✗ | 1.11s | 4783 |
| LCM-LoRA (Luo et al., 2023b)$^l$ | 1 | LoRA | 67.50 | 77.73 | 0.24 | ✗ | 0.11s | 1.3 |
| InstaFlow (Liu et al., 2023) | 1 | U-Net | 860M | 13.10 | 0.28 | ✗ | 0.11s | 183.2 |
| UFOGen (Xu et al., 2024) | 1 | U-Net | 860M | 12.78 | - | ✗ | - | - |
| DMD (Yin et al., 2024b) | 1 | U-Net | 860M | 11.49 | **0.32** | ✗ | 0.11s | 108 |
| DMD2 (Yin et al., 2024a)* | 1 | U-Net | 860M | 12.96 | 0.30 | ✓ | 0.11s | 5.1 |
| SiD-LSG (Zhou et al., 2024a)* | 1 | U-Net | 860M | 14.27 | 0.30 | ✓ | 0.11s | 6.4 |
| PCM (Wang et al., 2024) | 1 | U-Net | 860M | 17.91 | 0.29 | ✗ | - | unk |
| Hyper-SD (Ren et al., 2024)$^l$ | 1 | LoRA | 67.25M | 22.90 | 0.31 | ✗ | 0.11s | 33.3 |
| YOSO (Luo et al., 2024)$^l$ | 1 | LoRA | 67.25M | 23.68 | 0.29 | ✗ | 0.11s | 20 |
| DKD | 1 | LoRaD | 83.80M | **10.79** | 0.31 | ✓ | 0.11s | 2.1 |
| Stable Diffusion 2.1-based backbone | | | | | | | | |
| SD 2.1 (*cfg* = 3.0) (Rombach et al., 2022) | 25 | U-Net | 865M | 9.60 | 0.32 | ✗ | 1.04s | 8332 |
| SD-Turbo (Sauer et al., 2024)$^l$ | 1 | U-Net | 865M | 16.14 | **0.33** | ✗ | 0.11s | - |
| Swiftbrush (Nguyen & Tran, 2024) | 1 | U-Net | 865M | 16.67 | 0.29 | ✓ | 0.11s | 4.1 |
| Swiftbrushv2 (Dao et al., 2024)* | 1 | U-Net+LoRA | 884.14M | 15.98 | **0.33** | ✓ | 0.11s | 24.1 |
| SiD-LSG (Zhou et al., 2024a)* | 1 | U-Net | 865M | 15.17 | 0.30 | ✓ | 0.11s | 6.4 |
| TiUE (Li et al., 2025)$^l$ | 1 | U-Net | 865M | 13.49 | 0.31 | ✓ | 0.16s | 3.9 |
| DKD | 1 | LoRaD | 94.43M | 12.34 | 0.31 | ✓ | 0.11s | 2.1 |
| PixArt-$\alpha$-based backbone $256 \times 256$ | | | | | | | | |
| PixArt-$\alpha$ (*cfg* = 4.5) (Chen et al., 2023)$^l$ | 20 | DiT | 0.6B | 8.75 | 0.32 | ✗ | 0.59s | 753 |
| Swiftbrush (Nguyen & Tran, 2024)* | 1 | DiT | 0.6B | 29.89 | 0.28 | ✓ | 0.05s | 2.6 |
| PG-SB (Nguyen et al., 2024)* | 1 | DiT | 0.6B | 25.58 | 0.28 | ✓ | 0.05s | 2.6 |
| DKD | 1 | LoRaD | 81.22M | **18.99** | 0.30 | ✓ | 0.05s | 1.6 |

We compared the performance of DKD with other state-of-the-art (SOTA) methods on the zero-shot benchmark of COCO 2014. As shown in Tab. 5, DKD demonstrates excellent inference efficiency, with an inference time of 0.11 seconds on both the SD 1.5 and SD 2.1 backbones, making it one of the fastest methods. In terms of training, DKD completes distillation in only 2.1 A100 GPU days, significantly outperforming methods like InstaFlow (Liu et al., 2023) and DMD (Yin et al., 2024b), which require much longer training times. DKD achieves SOTA FID (10.79) and competitive CLIP score (0.31), striking a strong balance between speed and performance, particularly in image-free settings. This makes DKD an efficient distillation solution, especially in environments with limited computational resources.

We measured the inference time of the models on a server equipped with an NVIDIA A40 GPU, using a batch size of 1. The experiments were conducted with PyTorch 2.4.0 and Hugging Face

Please select the result that matches "the mona lisa" and has the best quality.

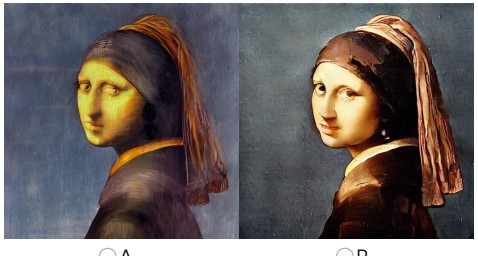

○A      ○B

Please select the result that matches "Dog inside bucket" and has the best quality.

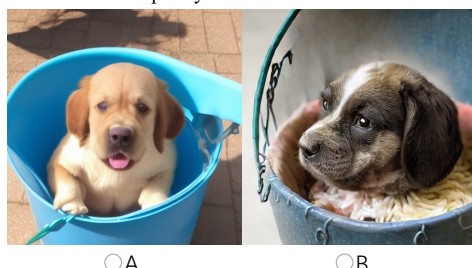

○A      ○B

Figure 15: User study examples.

Table 6: Quantitative comparison of DKD and other methods on HPSv2 results. * indicates our reproduced results, and $^l$ indicates results using the official pre-trained models. '-' denotes unknown. Best and second-best scores are in **bold** and underline, respectively.

| Method | Anime | Photo | Concept Art | Paintings | Average |
|---|---|---|---|---|---|
| Stable Diffusion 1.5-based backbone | | | | | |
| SD 1.5 (Rombach et al., 2022) | 26.51 | 27.19 | 26.06 | 26.12 | 26.47 |
| InstaFlow (Liu et al., 2023) | 26.10 | 26.62 | 25.92 | 25.95 | 26.15 |
| DMD2 (Yin et al., 2024a)* | 25.65 | 26.13 | 24.98 | 25.22 | 25.49 |
| SiD-LSG (Zhou et al., 2024a)* | 26.24 | 26.46 | 25.88 | 25.86 | 26.11 |
| Hyper-SD (Ren et al., 2024)$^l$ | **27.37** | **27.59** | **27.13** | **27.15** | **27.31** |
| YOSO (Luo et al., 2024)$^l$ | 26.24 | 26.26 | 25.92 | 25.79 | 26.05 |
| DKD | 26.39 | 26.80 | 25.79 | 25.81 | 26.20 |

Diffusers 0.25.0, with the inference time including the computation of the text encoder and latent decoder.

### F.5 USER STUDY DETAILS

This study recruited 57 volunteers from our university to participate in a questionnaire-based evaluation. The questionnaire consisted of 44 questions, each presenting several images—one generated by our DKD method and the others by alternative approaches, including SiD-LSG (Zhou et al., 2024a), DMD2 (Yin et al., 2024a), Hyper-SD (Ren et al., 2024), YOSO (Luo et al., 2024), SwiftBrush (Nguyen & Tran, 2024), SwiftBrushv2 (Dao et al., 2024), ControlNet (Zhang et al., 2023), Reversion (Huang et al., 2024b), DreamBooth (Ruiz et al., 2023), LoRA (Hu et al., 2022), and ScaleCrafter (He et al., 2023). An example of the questionnaire is shown in Fig. 15.

## G ADDITIONAL RESULTS

### G.1 RESULTS FOR HPSV2

Table 6 presents a quantitative comparison of DKD with other SOTA methods on the HPSv2 benchmark. With an average score of 26.20, DKD outperforms several competitive methods, including SiD-LSG (Zhou et al., 2024a) (26.11) and YOSO (Luo et al., 2024) (26.05). Notably, DKD excels in the photo category, achieving a score of 26.80, demonstrating its strong text-to-image alignment capability. It is worth noting that Hyper-SD (Ren et al., 2024) achieves SOTA performance across all metrics, thanks to its use of the aesthetic predictor of LAION dataset, the ImageReward (Xu et al., 2023) aesthetic preference reward model, and the SOLO (Wang et al., 2020) visual perception model, which guide the optimization process through multiple supervision signals.

Table 7: Ablation experiments on the impact of LoRaD application layer types.

| Type | #Trainable Params | FID | CLIP |
|------|-------------------|-----|------|
| Linear | 83.8M | **10.79** | **0.31** |
| Linear + Conv | 174.89M | 16.42 | 0.30 |

Table 8: Ablation experiments on LoraD initialization strategy.

| Initialization | FID | CLIP |
|----------------|-----|------|
| $A = 0$, $B = $ Xavier | **10.79** | **0.31** |
| $A = $ Xavier, $B = $ Xavier | 18.41 | 0.31 |

## G.2 ADDITIONAL ABLATION STUDIES

In our main experiments, we apply LoRaD only to linear layers to achieve a better trade-off between performance and parameter efficiency. As shown in Tab. 7, extending LoRaD to convolutional layers leads to a performance drop, suggesting that LoRaD already possesses sufficient representational capacity. Extending to more layers increases parameter count and may introduce overfitting.

To ensure the student initially matches the teacher, we initialize the student network with teacher weights. Specifically, in LoRaD, we set the low-rank matrix $A = 0$ and initialize $B$ with Xavier, resulting in $AB = 0$ at the start of training—thus applying no rotation and preserving the teacher's parameter directions. As shwon in Tab. 7, we also experimented with Xavier initialization for both $A$ and $B$, which led to a significantly worse FID ($10.79 \rightarrow 18.41$), indicating degraded convergence. This may be because non-zero initialization perturbs the pretrained model and causes optimization to converge to a suboptimal region, thereby affecting final performance.

Our zero initialization follows recent work. For example, both LoRA (Hu et al., 2022) and DoRA (Liu et al., 2024) set at the beginning of training, making the model initially equivalent to the pretrained weights , thus avoiding disruption of the original model behavior. Similarly, ControlNet Zhang et al., 2023 initializes the image-conditional branch to output zero, ensuring that the initial behavior remains consistent with the base model and allowing conditional control signals to be gradually introduced through training. This prevents training instability or performance degradation caused by the premature influence of untrained control branches. We believe that pretrained weights provide a strong anchor for distillation models, making optimization more stable and convergence easier. In addition, recent work PiSSA (Meng et al., 2024) is the first to apply SVD to the original model, leveraging principal singular values and vectors to initialize the adapter for fine-tuning. This approach further accelerates LoRA's convergence and improves its performance. These observations collectively highlight the crucial role of good initialization in achieving both fast convergence and strong final performance. Motivated by this, we plan to further investigate initialization strategies for LoRaD in future work.

## G.3 MORE QUANTITATIVE RESULTS

We compare DKD with representative distillation methods on COCO2014 (Lin et al., 2014), and further evaluate its generalization on COCO2017 (Lin et al., 2014). As shown in Tab. 9, DKD consistently achieves strong performance across three backbone models: SD 1.5, SD 2.1, and PixArt-$\alpha$. Specifically, DKD achieves the best or second-best results in both FID and CLIP scores across all settings, and strikes a favorable balance between precision and recall, demonstrating strong capability in image quality and semantic alignment. Notably, DKD is distilled using only 1.4M text prompts, yet outperforms or matches methods that rely on over 3M prompts, such as LCM (Luo et al., 2023b) (12M) and YOSO (Luo et al., 2024) (4M). This highlights both the efficiency of DKD under low-resource settings and the effectiveness of its distillation mechanism. In contrast to methods like Hyper-SD (Ren et al., 2024), YOSO (Luo et al., 2024), and SD-Turbo (Sauer et al., 2024), DKD requires no real images and is trained via prompt-only distillation, enhancing its practicality and scalability.

Table 9: Quantitative comparison of DKD and other methods on zero-shot COCO 2017 results. $^*$ indicates our reproduced results, and $^l$ indicates results using the official pre-trained models. '-' denotes unknown. Best and second-best scores are in **bold** and underline, respectively.

| Method | #Params | NFEs | Type | Trainable params | FID ↓ | CLIP ↑ | Precision ↑ | Recall ↑ | Image-free? | Training Data |
|---|---|---|---|---|---|---|---|---|---|---|
| *Stable Diffusion 1.5-based backbone* | | | | | | | | | | |
| SD 1.5 (*cfg* = 3.0) (Rombach et al., 2022) | 860M | 25 | U-Net | 860M | 19.80 | 0.31 | 0.64 | 0.60 | ✗ | 5B |
| LCM-LoRA (Luo et al., 2023b)$^l$ | 860M | 1 | LoRA | 67.50M | 89.65 | 0.24 | 0.22 | 0.24 | ✗ | 12M |
| InstaFlow (Liu et al., 2023) | 860M | 1 | U-Net | 860M | 23.49 | **0.31** | 0.53 | 0.46 | ✗ | 3.2M |
| UFOGen (Xu et al., 2024) | 860M | 1 | U-Net | 860M | 22.50 | **0.31** | - | - | ✗ | 12M |
| DMD2 (Yin et al., 2024a)$^*$ | 860M | 1 | U-Net | 860M | 23.30 | 0.30 | 0.60 | 0.49 | ✓ | 1.4M |
| SiD-LSG (Zhou et al., 2024a)$^*$ | 860M | 1 | U-Net | 860M | 24.22 | 0.30 | 0.60 | 0.52 | ✓ | 1.4M |
| Hyper-SD (Ren et al., 2024)$^l$ | 860M | 1 | LoRA | 67.25M | 32.49 | **0.31** | 0.52 | 0.33 | ✗ | - |
| YOSO (Luo et al., 2024)$^l$ | 860M | 1 | LoRA | 67.25M | 33.54 | 0.29 | 0.50 | 0.44 | ✗ | 4M |
| DKD | 860M | 1 | LoRaD | 83.80M | **20.86** | **0.31** | **0.63** | **0.54** | ✓ | 1.4M |
| *Stable Diffusion 2.1-based backbone* | | | | | | | | | | |
| SD 2.1 (*cfg* = 3.0) (Rombach et al., 2022) | 865M | 25 | U-Net | 865M | 19.66 | 0.32 | 0.66 | 0.57 | ✗ | 5B |
| SD-Turbo (Sauer et al., 2024)$^l$ | 865M | 1 | U-Net | 865M | 26.36 | **0.34** | **0.69** | 0.47 | ✗ | - |
| Swiftbrush (Nguyen & Tran, 2024) | 865M | 1 | U-Net | 865M | 26.87 | 0.32 | 0.61 | 0.44 | ✓ | 1.4M |
| Swiftbrushv2 (Dao et al., 2024)$^*$ | 865M | 1 | U-Net+LoRA | 884.14M | 25.96 | 0.33 | 0.65 | 0.45 | ✓ | 3.3M |
| SiD-LSG (Zhou et al., 2024a)$^*$ | 865M | 1 | U-Net | 865M | 25.02 | 0.30 | 0.62 | 0.51 | ✓ | 1.4M |
| DKD | 865M | 1 | LoRaD | 94.43M | **22.62** | 0.31 | 0.65 | **0.53** | ✓ | 1.4M |
| *PixArt-α-based backbone* | | | | | | | | | | |
| PixArt-α (*cfg* = 4.5) (Chen et al., 2023)$^l$ | 0.6B | 20 | DiT | 0.6B | 20.85 | 0.27 | 0.65 | 0.59 | ✗ | 25M |
| Swiftbrush (Nguyen & Tran, 2024)$^*$ | 0.6B | 1 | DiT | 0.6B | 41.07 | 0.28 | 0.53 | 0.35 | ✓ | 1.4M |
| PG-SB (Nguyen et al., 2024)$^*$ | 0.6B | 1 | DiT | 0.6B | 35.84 | 0.28 | 0.57 | 0.36 | ✓ | 1.4M |
| DKD | 0.6B | 1 | LoRaD | 81.22M | **28.91** | **0.30** | **0.62** | **0.37** | ✓ | 1.4M |

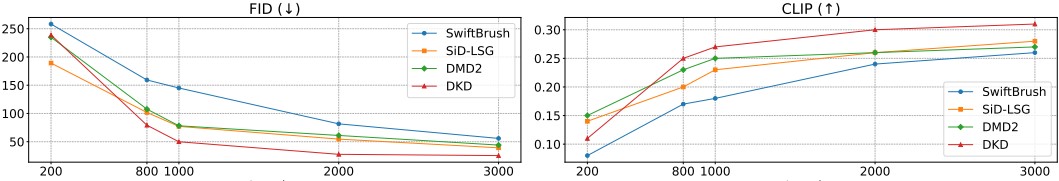

Figure 16: Convergence analysis of DKD and other methods. $^*$ indicates our reproduced results.

## G.4 CONVERGENCE ANALYSIS

Fig 16 presents the convergence analysis of DKD compared to other SOTA one-step distillation models, including SwiftBrush (Nguyen & Tran, 2024), SiD-LSG (Zhou et al., 2024a), and DMD2 (Yin et al., 2024a). The plot shows that DKD achieves faster convergence, with both FID and CLIP scores improving consistently across iterations. DKD demonstrates superior performance in terms of FID reduction, reaching a lower value than the other models by the end of the training. In terms of CLIP, DKD maintains a steady and significant increase, outperforming SwiftBrush (Nguyen & Tran, 2024) and SiD-LSG (Zhou et al., 2024a) in the later stages. This highlights DKD's efficiency in both training stability and perceptual alignment, which aligns with the qualitative results in Fig. 17.

## G.5 MORE VISUALIZATION RESULTS

Fig. 18 to Fig. 25 present the sampling results of DKD (based on SD 1.5), the sampling results of DKD (based on PixArt-α), additional qualitative comparisons, visualizations of ControlNet-DKD and Reversion-DKD, qualitative results of DreamBooth-LoRaD, and extended visualizations of ScaleCrafter-DKD, further demonstrating the generality and adaptability of our approach across diverse tasks.

|  | 200 | 800 | 1000 | 2000 | 3000 |
|---|---|---|---|---|---|

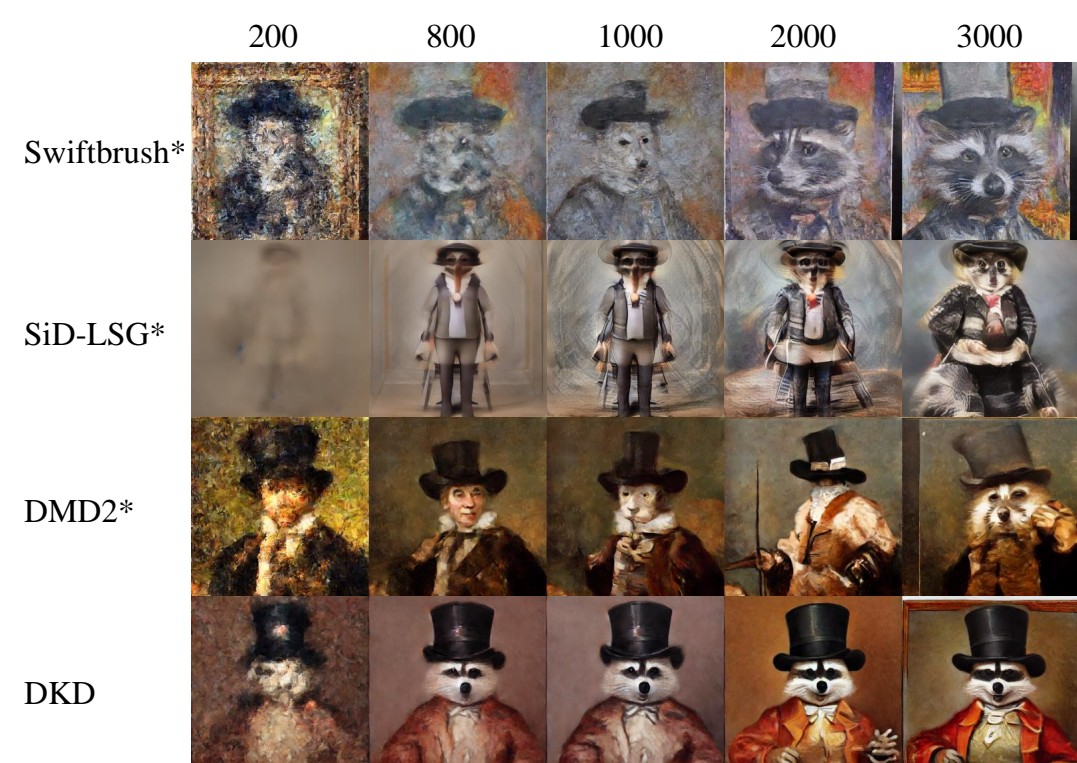

Swiftbrush*

SiD-LSG*

DMD2*

DKD

"A racoon wearing formal clothes, wearing a tophat. Oil painting in the style of Rembrandt"

Figure 17: Iteration qualitative results. "* indicates our reproduced results"

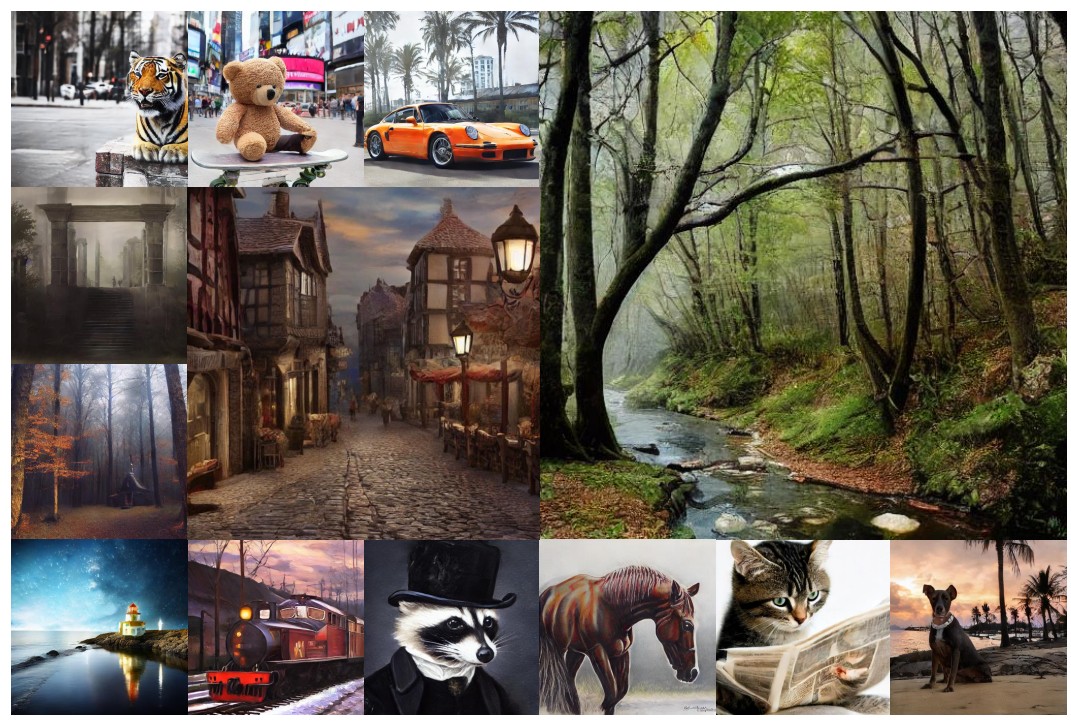

Figure 18: One-step generated images using our proposed method DKD (SD 1.5).

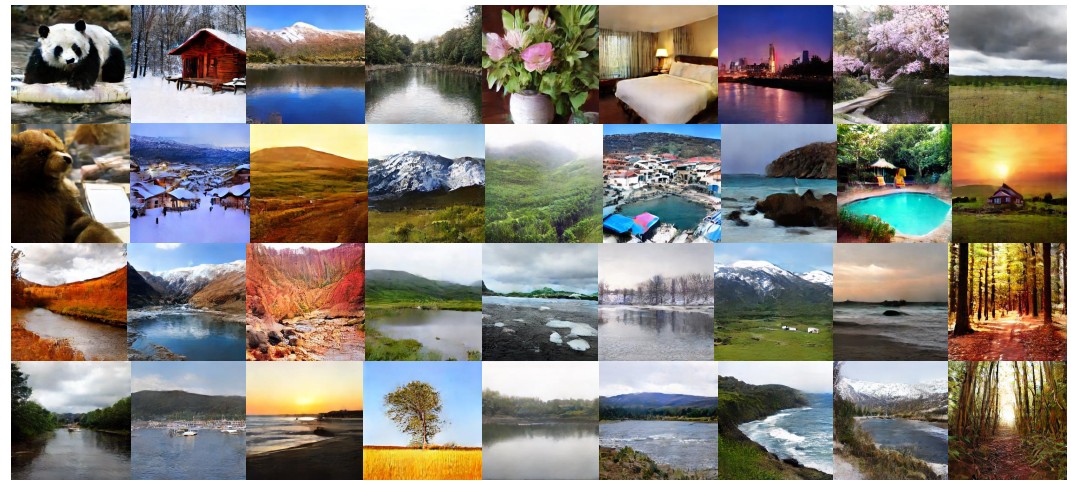

Figure 19: One-step generated images using our proposed method DKD (PixArt-$\alpha$ $256 \times 256$).

| SD 1.5-based Methods | SD 2.1-based Methods |
|---|---|
| LCM-LoRA   InstaFlow   DKD | SD-Turbo   TiUE   DKD |

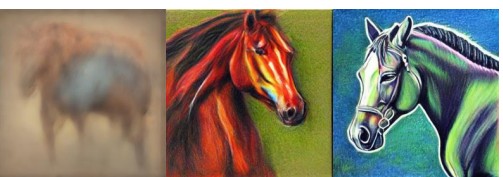

A hyperrealistic photo of a fox
astronaut, perfect face, artstation

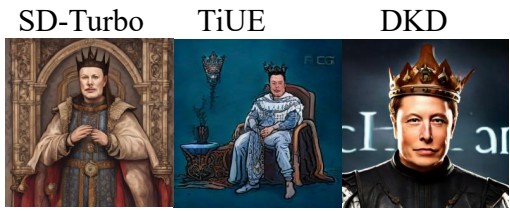

Elon Musk dressed as a medieval-style
king.

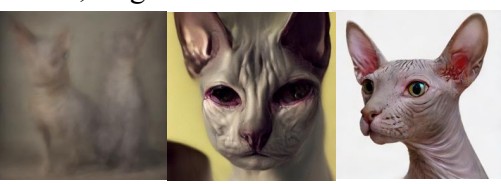

Masterpiece color pencil drawing of a
horse; bright vivid color

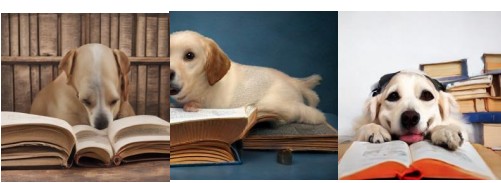

A dog is reading a thick book.

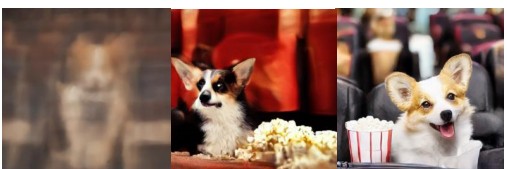

Highly detailed mysterious egyptian
(sphynx cat), skindentation: 1.2

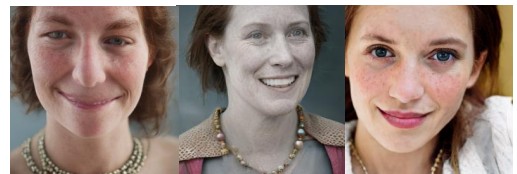

Portrait of a woman with freckles and
a necklace on her neck lightly smiling
at the camera

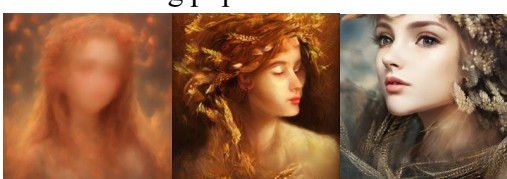

Cute small Corgi sitting in a movie
theater eating popcorn

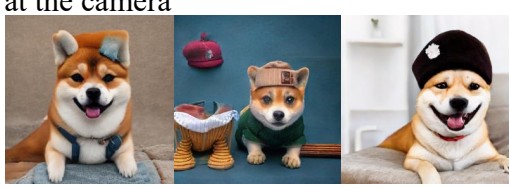

a shiba inu wearing a beret.

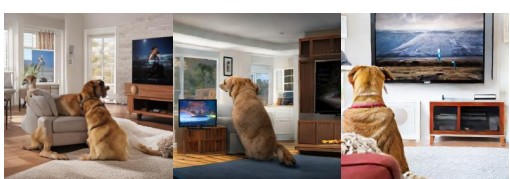

Large dog looking at television show
in living room.

Half-length head portrait of the
goddess of autumn with wheat ears
on her head, depicted as dreamy and
beautiful,by wlop

Figure 20: Qualitative comparison with other methods.

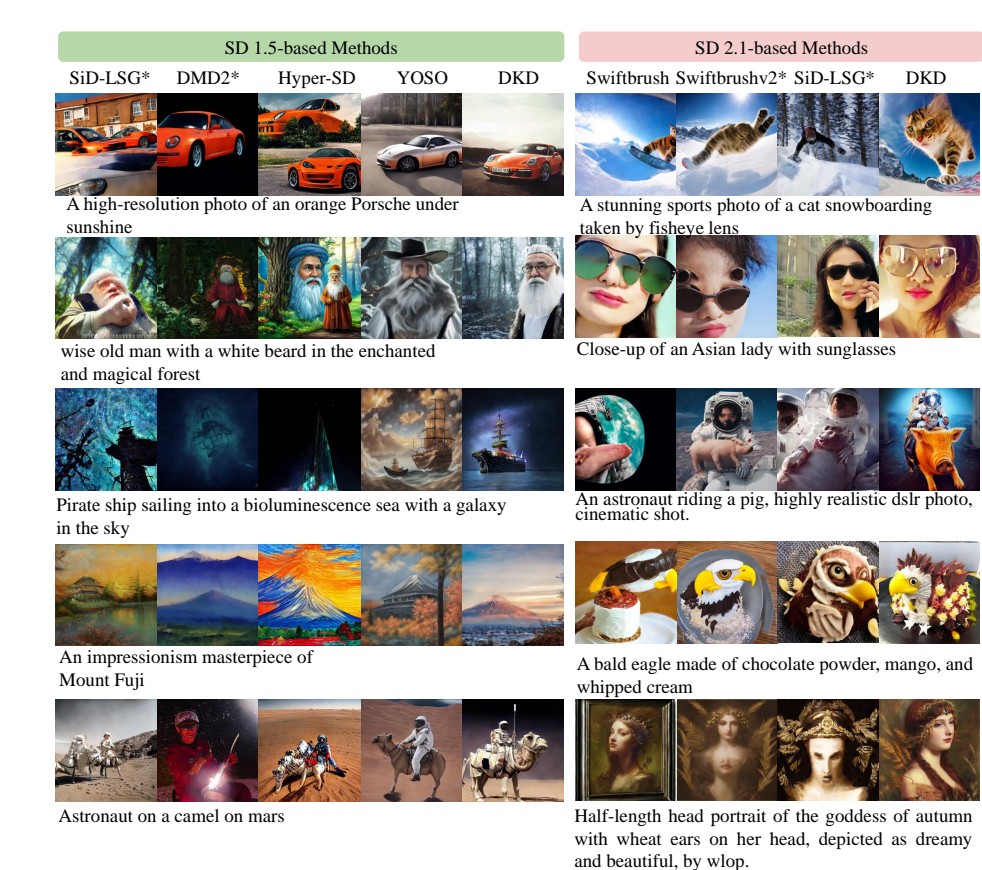

Figure 21: Qualitative comparison to state-of-the-art one-step distillation models. * indicates our reproduced results.

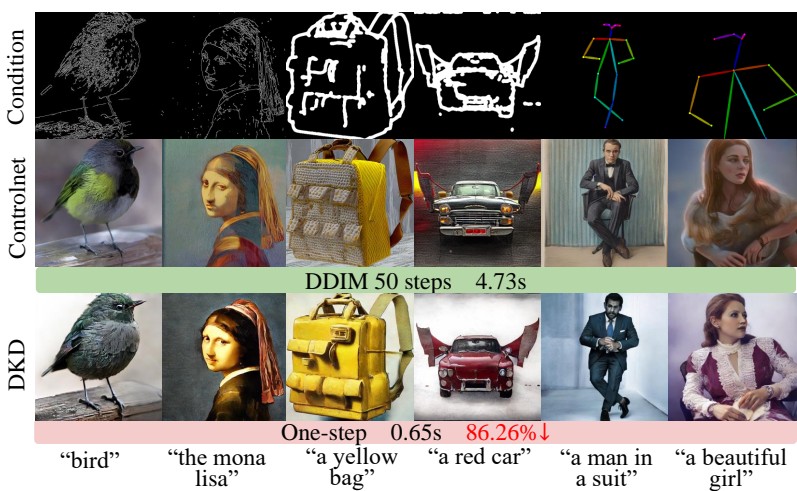

Figure 22: Quality results by Controlnet (Zhang & Chen, 2022) with or without DKD.

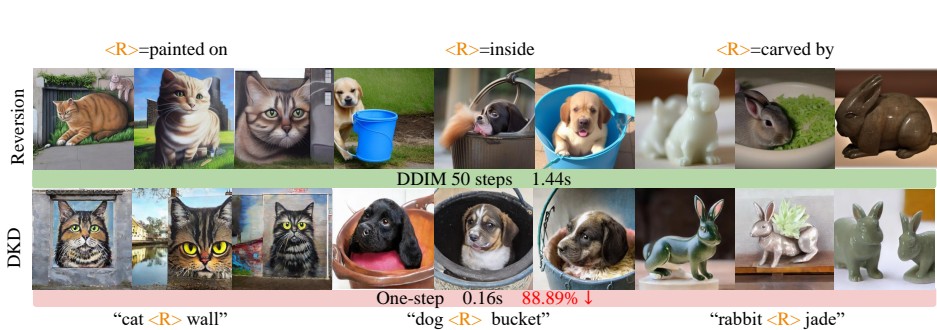

Figure 23: Quality results by Reversion (Huang et al., 2024b) with or without DKD.

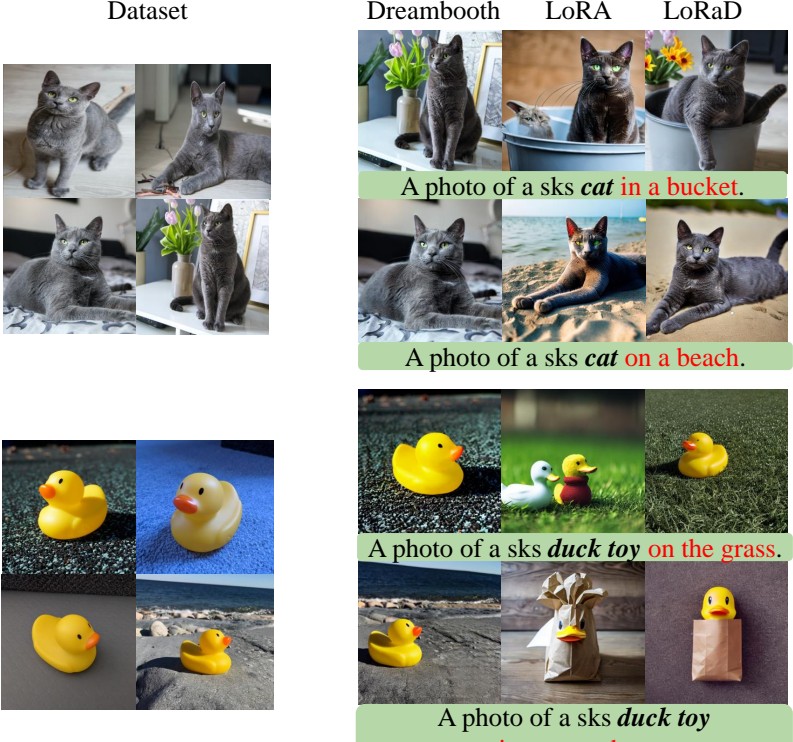

Figure 24: Quality results by Dreambooth (Ruiz et al., 2023) with or without LoRaD.

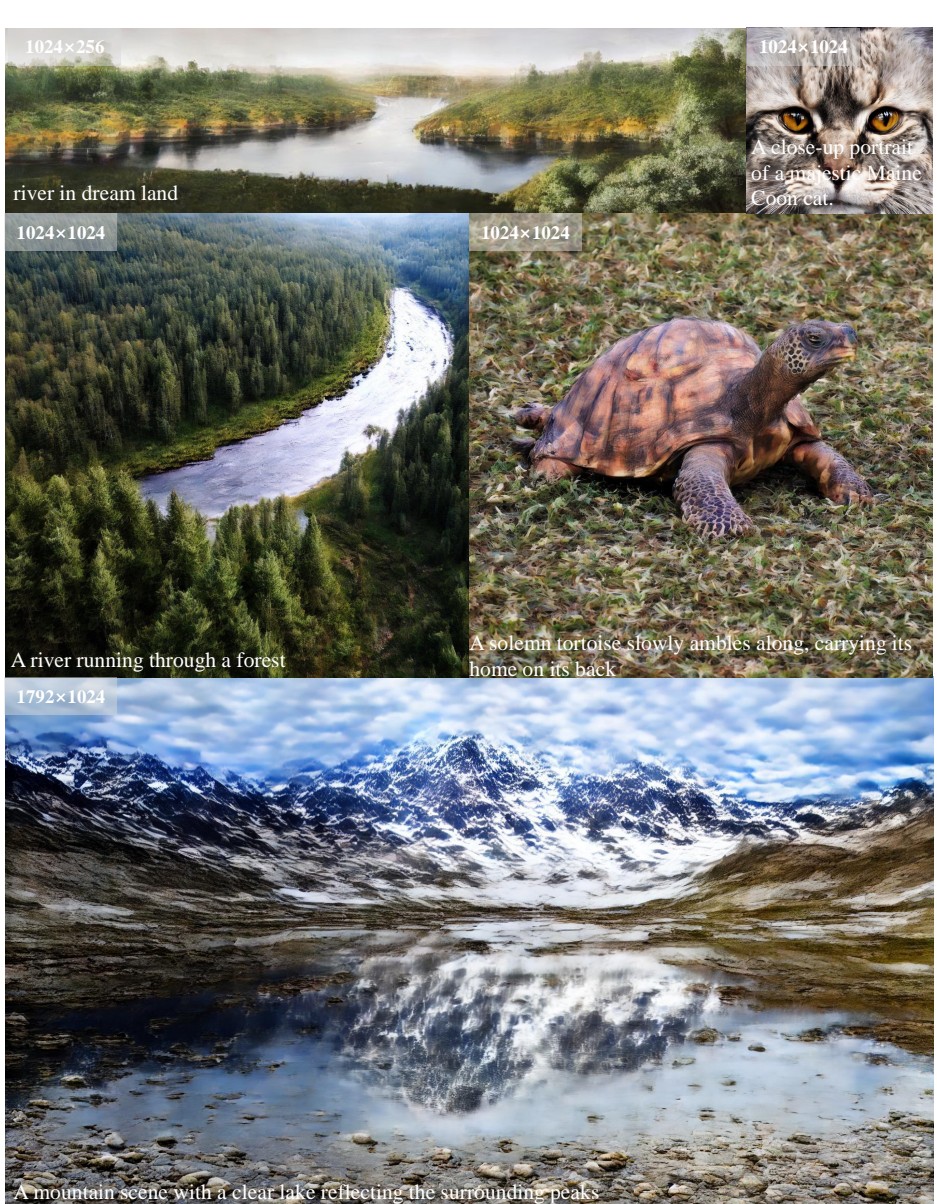

Figure 25: Quality results by ScaleCrafter (He et al., 2023) with DKD.

