# OpenReview forum: "DKD: Directional Knowledge Distillation for One-Step Text-to-Image Generation"
_ICLR.cc/2026/Conference — ICLR 2026 Conference Withdrawn Submission_

### Official Review · Reviewer_PK94 · 2025-10-28

**Soundness:** 1
**Presentation:** 2
**Contribution:** 1
**Rating:** 2
**Confidence:** 5

**Summary:**

The paper notices that T2I diffusion models and their few-step distilled counterparts exhibit significantly different weight directions while maintaining similar weight norms. Motivated by this insight, the authors present LoRaD, a low-rank adapter, parameterized with a learnable low-rank rotation matrix, to update only the directional component of the teacher’s weights. DKD incorporates LoRaD into the DMD method and is evaluated against various existing distillation approaches.

**Strengths:**

* Interesting analysis on the weight differences between the teacher and student models

**Weaknesses:**

* The proposed LoRaD design is not sufficiently justified. DoRA with frozen magnitudes seems as a suitable option to leverage the insight and it remains unclear why parameterization with a rotation matrix offers a superior alternative. More comprehensive empirical and theoretical investigation would be highly valuable.

* DKD itself is simply DMD with the proposed low-rank adapters. Therefore, it should not be presented as a new distillation method and I believe the comparisons against other publicly-available distilled models are less informative, given their different training setups. A more meaningful evaluation would be assessing LoRaD across multiple distillation methods under controlled settings. This would better highlight the LoRaD contribution to diffusion distillation.

* The main observation in Figure 1 does not generalize across different distilled models. For example, Figures 10–14 show that SwiftBrush and SD-Turbo have large stds in weight norms, making it difficult to conclude that weight norms are irrelevant in general.

**Evaluation**

* The comparisons rely solely on FID and CLIP scores. FID is widely recognized as an unreliable metric for T2I evaluation, especially on small (e.g., 5K) subsets. Precision and recall are also not widely adopted metrics for evaluating T2I models. Moreover, DKD performs comparably to or worse than baselines on CLIP and HPSv2 metrics.

* The ablation study in Table 2 is highly important, as it directly isolates the performance gains attributed to LoRaD. However, it should include a comparison with DoRA using frozen weight norms, more additional metrics (e.g, HPSv2, PickScore, ImageReward), and, perfectly, human studies.

### Minor weaknesses
* Paper structure: The analysis in Figure 1 in the introduction would be better placed in the analysis section following the related work.
* The user study figure is hard to read: multiple methods share a single bar making unclear which method performs best

**Questions:**

Please address the concerns in Weaknesses.

---

### Official Review · Reviewer_ThMX · 2025-10-29

**Soundness:** 1
**Presentation:** 1
**Contribution:** 2
**Rating:** 2
**Confidence:** 5

**Summary:**

The paper proposes Directional Knowledge Distillation (DKD), a one-step text-to-image distillation framework. The authors observe that changes in weight direction play a more critical role than changes in norm during distillation. Building on this, they introduce the Low-Rank Rotation of Weight Direction (LoRaD) module to efficiently model directional updates.

While the analysis is interesting, the technical contribution is limited. Specifically, the proposed parameterization, which is derived from the analysis, does not convincingly demonstrate clear benefits compared to the basic LoRA approach. Overall, I have a rather negative assessment of the paper, which requires significant revision.

**Strengths:**

* The paper considers different applications, including controllable generation, inversion, and image customization.
* The authors provide meaningful ablation studies.

**Weaknesses:**

* **Limited improvements over baselines.** Based on Table 2, the performance gap between LoRA/DoRA and LoRaD is minimal according to CLIP (0.31 vs 0.29/0.30), raising doubts about the effectiveness of the proposed rotation parameterization.
I acknowledge the improvements according to FID, but this is not the most reliable metric for evaluating text-to-image models *. Since these baselines are of greatest interest, the authors should include more diverse models (beyond SD1.5) and additional metrics (e.g., human evaluation, ImageReward, PickScore) to provide stronger evidence for the importance of the proposed parameterization. Moreover, more distillation methods, such as consistency distillation, should be considered.

*SDXL:Improving Latent Diffusion Models for High-Resolution Image Synthesis.


* **Outdated experimental setup.** The paper evaluates outdated text-to-image models (SD1.5/2.1), which weakens the empirical claims. To provide a convincing demonstration, the method should be tested on contemporary backbones (e.g., SD3/3.5 or FLUX). At minimum, SDXL should be included, along with comparisons against strong open-source models such as DMD2.


* **Poor presentation.** The paper is not well written, and several presentation issues hinder readability. For example, Figure 2 is visually overwhelming and contains typos (e.g., “(d) Qualitative examples corresponding to (b)”).

**Questions:**

* Which layers are most important for including LoRaD? I found the analysis for linear/convolutional layers in Table 7, but what about transformer backbones?
* Can the proposed parameterization be applied to other distillation approaches (e.g., consistency distillation), and what benefits could it potentially bring?

---

### Official Review · Reviewer_TJGj · 2025-10-30

**Soundness:** 3
**Presentation:** 3
**Contribution:** 2
**Rating:** 4
**Confidence:** 2

**Summary:**

The submission compares the weights' changes between multi-step and one-step diffusion models and reveals that changes mostly happen in the direction of weights rather than norms, and show that these direction changes are low-rank. Based on these observations the authors propose to update only the direction component of the weight matrix and implement this logic via the specific parametrization of learnable weights.

**Strengths:**

(1) The proposed method is motivated by the insightful analysis, which is interesting regardless of the practical benefits of the proposed distillation method.

(2) The advantage of the proposed parameterisation is demonstrated in the wide range of downstream applications (ControlNet, relation inversion, DreamBooth-like customization).

(3) The quantitive advantage of the proposed LoRAD parameterisation over the considered baselines is impressive (Table 1, Figure 9).

**Weaknesses:**

(1) I do not understand why the proposed parameterisation is positioned as a component of the distillation method. As demonstrated by the Dreambooth experiments, LoRAD can be beneficial outside the distillation context as well.

(2) It is not clear if the proposed LoRAD outperforms the existing natural baselines. For instance, in DoRA (ICML'2024) one could also freeze the magnitude component of the weights, but the authors do not compare with this baseline.

(3) It is not clear if the proposed LoRAD can be used for few-step generation to trade the runtime inference for generative quality. Why the positioning is restricted to one-step inference scenario?

**Questions:**

Please, address my concerns from the Weaknesses section.

---

### Official Review · Reviewer_QLiT · 2025-11-01

**Soundness:** 2
**Presentation:** 2
**Contribution:** 2
**Rating:** 4
**Confidence:** 4

**Summary:**

This paper reveals  the changes in weight direction are the leading variations between one-step students and their multi-step teacher models based on latent diffusion models.  Motivated
by this finding, the authors propose Low-rank Rotation of weight Direction (LoRaD) to learn the structured directional changes.  DKD, through integrating LoRaD into VSD,  achieves state-of-the-art FID scores on
COCO 2014 and COCO 2017.

**Strengths:**

1. This paper conducts an in-depth analysis of weight changes in the U-Net between multi-step and one-step generation models, identifying weight-direction adjustment as the key driver of one-step distillation and thereby establishing a novel theoretical foundation for efficient distillation.

2. DKD introduces a novel distillation paradigm for one-step text-to-image generation， which  leverages LoRaD to model weight directions via low-rank rotations, ensuring precise alignment of the student model with the teacher distribution.

3. This paper conducts experiments on COCO uisng SD1.5, SD2.1, and PixArt-α. The results demonstrate that DKD improves inference efficiency while achieving substantial gains in image quality

**Weaknesses:**

1. This paper  presents LoRaD as a general method, but it only valiates its effectiveness for the distillation of latent diffusion models (LDM), lacking the finetuning results of  LDM and MLLM. I suggest that the authors describe  LoRaD in the perspective of the accelerating LDMs.

2. This paper only shows the results on small  LDMs, making the conclusion not convincing. The finding of weight changes is also true on SDXL and SD3 or Flex-dev.1？ Therefore， the results on  SDXL and SD3 or Flex-dev.1 should be presented.

3. As DKD integrates LoRaD into VSD for model distillation, I can not recognize the improvement come from which component.  Is LoRaD or VSD is the core contribution to the performance gain. How about directly combining LoRaD  to DMD, DMD2, PCM. Such comparisons are more clear to reflect the advantage of LoRaD than LoRA.

**Questions:**

This lacks human preference metric to evaluate the model performance. ImageReward, HPSv2, and MPS[1] need to be included in the paper.

Reference

1. Learning multi-dimensional human preference for text-to-image generation.

---

### Note · Authors · 2025-11-14

I have read and agree with the venue's withdrawal policy on behalf of myself and my co-authors.